



**Predictive mapping of organic carbon stocks and accumulation rates**
**in surficial sediments of the Canadian continental margin**
Graham Epstein[1*], Susanna D. Fuller[2], Dipti Hingmire[3], Paul G. Myers[4], Angelica Peña[5], Clark Pennelly[4]
& Julia K. Baum[1]
[1]Department of Biological Sciences, University of Victoria, Victoria, British Columbia, Canada, V8P 5C2
[2]Oceans North, Halifax, NS B3J 1E6, Canada.
[3]School of Earth and Ocean Sciences (SEOS), University of Victoria, Victoria, British Columbia, Canada,
V8P 5C2
[4] Department of Earth and Atmospheric Sciences, University of Alberta, Edmonton, Canada, AB T6G 2E3
[5] Institute of Ocean Sciences, Fisheries and Ocean Canada, Sidney, British Columbia, Canada, V8L 4B2
*Corresponding author: Email - grahamepstein@uvic.ca





**Abstract**
The quantification and mapping of surficial seabed sediment organic carbon has wide-scale
relevance across marine ecology, geology and environmental resource management, with carbon
densities and accumulation rates being a major indicator of geological history, ecological function,
and ecosystem service provisioning, including the potential to contribute to nature-based climate
change mitigation. While global mapping products can appear to provide a definitive
understanding of the spatial distribution of sediment carbon, there is inherently high uncertainty
when making estimates at this scale. Finer resolution national maps which utilise targeted data
syntheses and refined spatial data products are therefore vital to improve these estimates. Here,
we report a national systematic review of data on organic carbon content in seabed sediments
across Canada and combine this with a synthesis and unification of best available data on
sediment composition, seafloor morphology, hydrology, chemistry, geographic setting and
sediment mass accumulation rates within a machine learning mapping framework. Predictive
quantitative maps of mud content, sediment dry bulk density, and organic carbon content, density
and accumulation, were each produced along with cell specific estimates of their 95% confidence
interval (CI) bounds at 200 m resolution across 4,489,235 km$^2$ of the Canadian continental margin
(92.6% of the seafloor area above 2,500 m). Fine-scale variation in carbon stocks was identified
across the Canadian continental margin, particularly in the Pacific and Atlantic Ocean regions.
Carbon accumulation was predicted to be concentrated in coastal areas, with the highest rates in
the Gulf of St Lawrence and Bay of Fundy. Overall, we estimate the standing stock of organic
carbon in the top 30 cm of surficial seabed sediments across the Canadian shelf and slope to be
10.7 Gt (95% CI 6.6 – 16.0 Gt), and accumulation at 4.9 Mt per year (95% CI 2.6 – 9.3 Mt y$^{-1}$).
Increased *in-situ* sediment data collection and higher precision in spatial environmental data-
layers could significantly reduce uncertainty and increase accuracy in these products over time.
**1. Introduction**
The organic carbon contained in seafloor sediments has a major influence on global carbon cycles
and earth's climate (Hülse et al., 2017; Bauer et al., 2013). Seabed sediments have been
estimated to accumulate approximately 126–350 Mt of organic carbon per year (Keil, 2017;
Berner, 1982) and contain 87 Gt of organic carbon in their top 5 cm (Lee et al., 2019), 168 Gt in
the top 10 cm (LaRowe et al., 2020a) and up to ~2,300 Gt in the top 1 m (Atwood et al., 2020),
with the latter being equivalent to nearly twice that of soils on land. Continental shelves have the



highest concentrations of sediment carbon across the global ocean, covering only 5-8% of the
marine area but an estimated 15-19% of surficial organic carbon stocks (LaRowe et al., 2020a;
Atwood et al., 2020) and 80% of annual carbon burial (Bauer et al., 2013; Burdige, 2007).
Continental margin zones (continental shelves and slopes) also contain the largest spatial
variation in organic carbon densities due to highly heterogenous geological, geographic, biological
and oceanographic settings (Smeaton et al., 2021; Diesing et al., 2017, 2021; Atwood et al.,
2020). They are also subjected to high levels of human activity, being impacted by many coastal
and marine industries including fishing, shipping, energy generation, telecommunication, mineral
extraction, and pollution from land based activities  (Halpern et al., 2019; Amoroso et al., 2018;
Keil, 2017). The quantification and mapping of organic carbon on continental margins  is therefore
imperative for best practise seabed management; with the densities and accumulation rates being
a major indicator of ecological function, geological history and ecosystem service provision
(Legge et al., 2020; Snelgrove et al., 2018; Middelburg, 2018).
In the marine environment, organic carbon can originate from the fixation of carbon dioxide ($CO_2$)
by primary producers in the photic zone or via lateral transport from terrestrial sources (LaRowe
et al., 2020b). Organic carbon then passes through a variety of biotic and abiotic pathways being
consumed, transformed, respired or remineralised, with a large proportion converted back into
inorganic compounds, leaving only ~5% of marine production and less than 1% of earth's gross
production eventually reaching the seafloor (Middelburg, 2019; Hülse et al., 2017; Turner, 2015;
Bauer et al., 2013; Burdige, 2007). Once at the seafloor, a similarly complex process occurs on
and within the sediment, with a wide range of biotic, biochemical and physical processes all
influencing the rates of accumulation, remineralisation and resultant long term burial, with ~90%
of all carbon reaching the seafloor being remineralised (LaRowe et al., 2020b; Middelburg, 2018,
2019; Arndt et al., 2013). Even when considering this complex carbon cycle, the mass and
accumulation of organic carbon in surficial seabed sediments will still have a direct influence on
the scale of long-term carbon storage at the seafloor (LaRowe et al., 2020a; Middelburg, 2018).
Marine habitats are being increasingly recognised as contributors to nature-based climate change
mitigation (also known as nature-based climate solutions and natural climate solutions) due to
their ability to both fix $CO_2$ and store organic carbon for centennial to millennial timescales
(Macreadie et al., 2021; Hoegh-Guldberg et al., 2019). This "blue carbon" potential was initially
recognised in coastal vegetated habitats (i.e. mangrove, seagrass and saltmarsh) (Nellemann et
al., 2009; Duarte et al., 2005), but has more recently been applied to other habitats such as kelp
forests and unvegetated sediments (Luisetti et al., 2020; Raven, 2018; Avelar et al., 2017). There





is increasing evidence that human activities are influencing seabed sediment carbon stores from
both perturbations of upstream processes and physical impacts directly on the seafloor (Cavan
and Hill, 2022; Epstein et al., 2022; Keil, 2017; Bauer et al., 2013). For example, a recent study
estimated that the direct physical impacts from global fishing activities could cause considerable
remineralisation of seabed sediment organic carbon stocks back to $CO_2$ (Sala et al., 2021),
however the validity of the scale of these estimates has been called into question (Hiddink et al.,
2023; Hilborn and Kaiser, 2022; Epstein et al., 2022). By improving the accuracy in available
sediment carbon mapping products, there may be potential to better research and design
appropriate management strategies to enhance organic carbon accumulation or limit
remineralisation from disturbance (Epstein and Roberts, 2022; Sala et al., 2021; Luisetti et al.,
91   2019).

Historically, studies measuring seabed sediment carbon stocks and accumulation rates had small
geographic scope, largely considering the ecological function, geological characteristics or
biochemical functioning at local to regional scales (see citations within LaRowe et al., 2020b;
Snelgrove et al., 2018; Middelburg, 2018; Burdige, 2007). In recent years, made possible by
modern machine learning and statistical spatial prediction techniques, there has been increasing
interest in estimating the size and distribution of carbon standing stocks and accumulation rates
at national to global scales to better understand natural carbon cycles and biological productivity,
and to identify the potential for improved management as a natural climate mitigation strategy
(Restreppo et al., 2021; Smeaton et al., 2021; Diesing et al., 2021; Atwood et al., 2020; LaRowe
et al., 2020b; Lee et al., 2019; Wilson et al., 2018; Avelar et al., 2017). Although global mapping
products can appear to give a complete understanding of seabed sediment organic carbon stocks,
there is high inherent uncertainty when making estimates at this scale (Ludwig et al., 2023;
Atwood et al., 2020; Lee et al., 2019). This has been highlighted by several regional studies across
the northwest European shelf (Smeaton et al., 2021; Diesing et al., 2017, 2021; Luisetti et al.,
2020; Wilson et al., 2018), which show distinct spatial patterns in organic carbon distribution and
disparate estimates of total standing stocks when compared with these global studies.
Canada has the world's longest coastline and approximately the seventh largest Exclusive
Economic Zone (EEZ) (Fig. 1), it could therefore be expected to contain a significant proportion
of the global stock of seabed sediment organic carbon. Data from recent global studies estimated
that the Canadian EEZ contains approximately 2.2 Gt of organic carbon in the top 5 cm and 48
Gt in the top meter of seabed sediments, equivalent to ~2.3% of total global marine sediment
carbon stocks covering around 1.3% of the area (Atwood et al., 2020; Lee et al., 2019). However,



these modelled estimates from global studies are at coarse spatial resolutions, have incomplete
coverage and contain very limited *in-situ* data from within the Canadian EEZ itself. The Canadian
marine environment is extremely complex, covering three oceans, 46 degrees of latitude, 94
degrees of longitude, and containing numerous features including the largest enclosed marine
bay in the world, over 50,000 islands, and on the comparatively short Pacific coastline alone,
around 436 estuaries. It is therefore highly likely that global estimates of the distribution of seabed
sediment organic carbon stock and accumulation rates are inaccurate for this region, and a
national approach is needed. Here, we conduct a systematic review of data on seabed sediment
organic carbon content across Canada and combine this with a synthesis and unification of best
available data on sediment composition, seafloor morphology, hydrology, chemistry and sediment
mass accumulation rates in a machine learning predictive mapping process, to construct the first
national assessment of Canadian seabed sediment organic carbon stocks and accumulation
rates.

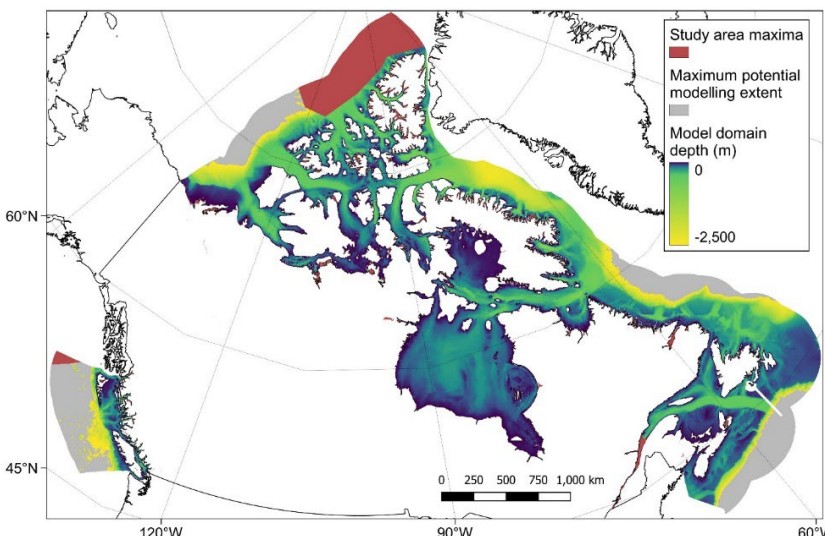


**Figure 1. Map of the Canadian Exclusive Economic Zone (EEZ).** The study area spatial maxima (red; see high
resolution figure for further detail around the coastline) covers the entire sub-tidal portion of the Canadian EEZ. This is
overlayed by the maximum potential modelling extent (grey) which only includes those areas where data were present
for all predictor variables. Due to the distribution of available data, the final model domain was limited to a depth of
2,500 meters, and is indicated with the colour relative to the estimated depth, from 0 (dark blue) to -2,500 (yellow).
Country       outlines      from      World      Bank      Official      Boundaries,      available      at
https://datacatalog.worldbank.org/search/dataset/0038272.



## 2. Methods

### 2.1 Analysis software

Analyses were primarily undertaken in R 4.2.2 (R Core Team, 2022) and RStudio 2022.12.0.353 (Posit Team, 2022), with some additional data manipulation and spatial plotting in QGIS (QGIS.org, 2021) and Python (Van Rossum and Drake, 2009). Within R, raster data were handled using the *terra* package (Hijmans, 2022), spatial vector data using the *sf* package (Pebesma, 2018), netCDF data with the *stars* (Pebesma, 2022) and *tidync* (Sumner, 2022) packages, data-frames with the *dplyr* package (Wickham et al., 2019), and vector data with base R (R Core Team, 2022). Random forest modelling was primarily dependent on the *ranger* package (Wright and Ziegler, 2017), however models were constructed and tuned using the *tidymodels* package (Kuhn and Wickham, 2020), with cross-validation and predictor variable selection using the *CAST* (Meyer et al., 2023) and *caret* (Kuhn, 2022) packages. Plotting utilised the above packages as well as *ggplot2* (Wickham et al., 2019) and *patchwork* (Pedersen, 2022) while parallel processing used the *doParallel* package (Microsoft Corporation and Weston, 2022). To aid clarity, a workflow diagram of the proceeding methods and results sections is shown in Figure 2.

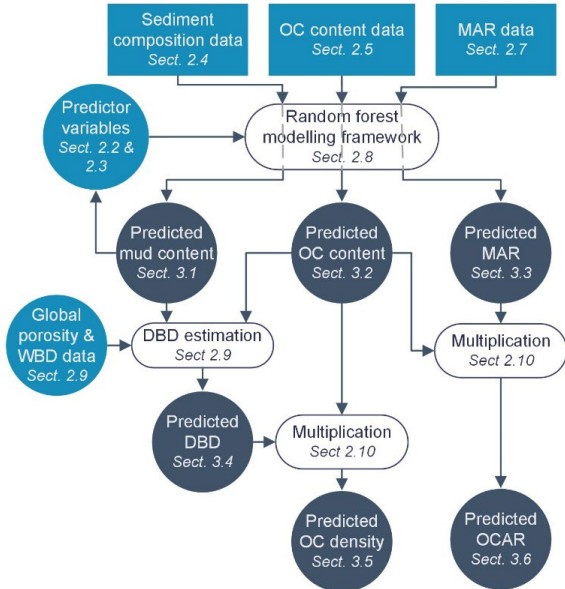

**Figure 2. Study workflow diagram.** Outline of the structure and linkages within the proceeding methods and results sections. Light blue shapes indicate input data; white ovals indicate data processes; dark shapes indicate output data; rectangles indicate point data; circles indicate raster data. OC = organic carbon; MAR = mass accumulation rate; WBD = wet bulk density; DBD = dry bulk density; OCAR = organic carbon accumulation rate.





**2.2 Study area spatial maxima**

To define the maximum potential spatial coverage of this study, best available bathymetric datasets were combined across the Canadian Exclusive Economic Zone (EEZ) (Table 1). Firstly, three Digital Elevation Model (DEM) raster layers covering different extents of the Canadian EEZ were each filtered to contain only those elevations of less than or equal to 0 m. Where necessary, data were then aggregated (averaged) or disaggregated (split) to a resolution of approximately 200 m, and all layers were projected onto a unified 200 m x 200 m equal area grid (co-ordinate reference system (CRS) EPSG:3573 - WGS 84 - North Pole Lambert Azimuthal Equal Area Canada). Reprojection was necessary as all three DEMs were in different co-ordinate systems, including some already being projected. The 200 m resolution was chosen as it is the median native resolution of the three DEMs, while also being considered towards the upper limit of what may be computationally possible within the scope of this study. After reprojection, the three layers were overlain, with the region-specific data given priority over global data where present. Finally, the seaward boundaries were delineated by the outer extent of the Canadian EEZ (Flanders Marine Institute, 2019). The resultant bathymetric layer was defined as the study area spatial maxima and used as the first potential predictor variable in predictive modelling (Fig. 1 – covering all coloured areas; Table 1).



**Table 1. Summary of predictor variables constructed for the Canadian EEZ.** For more information on methods
used to derive these layers see Sections 2.1 and 2.2.

| Predictor variable | Unit | Region | Source | Native resolution | Temporal range |
|---|---|---|---|---|---|
| Bathymetry | m | BC | NRCan (2021) | 10 m | NA |
| | | Arctic | IBCAO V4.2 (Jakobsson et al., 2020) | 200 m | NA |
| | | Global | GEBCO (2022) | 0.0042° | NA |
| Slope | ° | Canada | This study | 200 m | NA |
| Slope smoothed | ° | Canada | This study | 1 km | NA |
| Total curvature | rad/m | Canada | This study | 200 m | NA |
| Total curvature smoothed | rad/m | Canada | This study | 1 km | NA |
| BPI – fine | m | Canada | This study | 200 m | NA |
| BPI - medium | m | Canada | This study | 400 m | NA |
| BPI - broad | m | Canada | This study | 400 m | NA |
| VRM – fine | - | Canada | This study | 200 m | NA |
| VRM - medium | - | Canada | This study | 200 m | NA |
| VRM - broad | - | Canada | This study | 400 m | NA |
| Distance to shore | m | Canada | This study | 200 m | NA |
| Bioregion | - | Canada | DFO (2022) | NA | NA |
| Distance to rivers - large | m | Canada | NRCan (2019) | 1:15000000 | NA |
| Distance to rivers - medium | m | Canada | NRCan (2019) | 1:5000000 | NA |
| Distance to rivers - small | m | Canada | NRCan (2019) | 1:1000000 | NA |
| Exposure proxy | - | Canada | This study | 200 m | NA |
| SPM (surface) | g/m$^3$ | Global | Copernicus (2022b) | 4 km | 2007 - 2019 |
| Wave velocity (seafloor) | m/s | Arctic | Copernicus (2022a) | 3 km | 2007 - 2019 |
| | | Global | Copernicus (2022c) | 0.2° | 2007 - 2019 |
| Mean current velocity (seafloor) | m/s | BC | Peña et al. (2019) | 3 km | 2007 - 2019 |
| | | Salish Sea | SalishSeaCast ERDDAP v19-05* | 500 m | 2007 - 2019 |
| | | Arctic & Atlantic | ANHA12 (Hu et al., 2019)[†] | 0.0833° | 2007 - 2019 |
| Temperature (seafloor) | °C | BC | Peña et al. (2019) | 3 km | 2007 - 2019 |
| | °C | Salish Sea | SalishSeaCast ERDDAP v19-05* | 500 m | 2007 - 2019 |
| | °C | Arctic & Atlantic | ANHA12 (Hu et al., 2019)[†] | 0.0833° | 2007 - 2019 |
| Salinity (seafloor) | ppt | BC | Peña et al. (2019) | 3 km | 2007 - 2019 |
| | | Salish Sea | SalishSeaCast ERDDAP v19-05* | 500 m | 2007 - 2019 |
| | | Arctic & Atlantic | ANHA12 (Hu et al., 2019)[†] | 0.0833° | 2007 - 2019 |
| Ice thickness (surface) | m | Arctic & Atlantic | ANHA12 (Hu et al., 2019)[†] | 0.0833° | 2007 - 2019 |
| Ice concentration (surface) | % | Arctic & Atlantic | ANHA12 (Hu et al., 2019)[†] | 0.0833° | 2007 - 2019 |
| Dissolved oxygen (seafloor) | mol/m$^3$ | Global | Bio-ORACLE V2.2 (Assis et al., 2018) | 0.0833° | 2000 - 2014 |
| Primary production (surface) | g/m$^3$/d | Global | Bio-ORACLE V2.2 (Assis et al., 2018) | 0.0833° | 2000 - 2014 |
| Chlorophyll concentration (surface) | mg/m$^3$ | Global | Bio-ORACLE V2.2 (Assis et al., 2018) | 0.0833° | 2000 - 2014 |
| Max current velocity (seafloor) | m/s | Global | Bio-ORACLE V2.2 (Assis et al., 2018) | 0.0833° | 2000 - 2014 |

Notes: BC = British Columbia; BPI = Benthic position index; VRM = Vector ruggedness measure; SPM = Suspended
particulate matter. *See https://salishsea.eos.ubc.ca/erddap/index.html; Soontiens and Allen (2017); Soontiens et al.
(2016). [†]See: https://canadian-nemo-ocean-modelling-forum-community-of-
practice.readthedocs.io/en/latest/Institutions/UofA/Configurations/ANHA12/index.html




**2.3 Predictor variables**

2.3.1 Benthic terrain features

A set of 10 benthic terrain features were constructed from the unified bathymetric layer (Table 1). As benthic terrain measures use data on the depth of a location relative to the depth of surrounding cells up to a given distance, bathymetric data within a given buffer outside the study area maxima were included as needed to avoid edge effects in each terrain feature. Slope and total curvature were calculated using the *terra.terrain* (Hijmans, 2022) and *spatialEco.curvature* (Evans and Murphy, 2021) functions respectively. As these measures can be particularly sensitive to artifacts from the DEM models and projections, they were constructed at two resolutions – the native 200 m resolution, and after aggregating the bathymetry by 5-fold to 1 km x 1 km (termed "smoothed"). Smoothed layers were disaggregated back to a 200 m resolution to maintain uniformity across predictor layers.

Benthic position index (BPI) and vector ruggedness measures (VRM) were each calculated using the *MultiscaleDTM* package at 3 different levels to capture both small local features and larger spatial variation in terrain (Ilich et al., 2021). Benthic position index was calculated as the difference between the depth of a focal cell and the mean of cells contained in annulus shaped window of 0.2 km to 5 km (BPI fine), 2 km to 25 km (BPI medium) and 4 km to 100 km (BPI broad). Vector ruggedness was measured by considering variation in the depth surrounding each cell within square windows of width 1 km (VRM fine), 5.8 km (VRM medium) and 11.6 km (VRM broad). Due to extremely inhibitive computational times when calculating VRM broad, BPI medium and BPI broad at 200 m resolution, for these features the bathymetric layer was first aggregated to a 400 m resolution before feature calculation, and then disaggregated back to 200 m to maintain uniformity.

2.3.2 Predictors describing the geographic setting

The geographic setting of each cell was described by its distance to shore and rivers, its broad bioregional classification, and a proxy measure describing the degree of exposition vs. shelteredness (Table 1). The geographic setting features are also influenced by the values of surrounding pixels, therefore appropriate buffers were also applied to the processing of these layers to avoid edge effects. Distance to shore was measured by the Euclidian distance to the nearest land cell (indicated by an 'NA' value in the bathymetry layer), while bioregion was defined





by the Fisheries and Oceans Canada Federal Marine Bioregions classification (DFO, 2022). The
bioregion polygons were edited to include all bathymetry cells and re-classified with an integer
scale of 1 to 12 from east to west.
CanVec is a digital cartographic reference product produced by Natural Resources Canada
(NRCan) which includes the location of rivers across Canada at three mapped scales (NRCan,
2019). Firstly, the coarsest scale data (1:15,000,000) was projected onto the CRS of the
bathymetry layer and converted from polylines to a 2 km resolution raster. A 2 km buffer was
added around each river to ensure overlap of river mouths with the bathymetry data. The resultant
raster layer was resampled onto the bathymetry raster and the grid distance of each bathymetry
cell to the nearest river-mouth cell was calculated using the *terra.gridDist* function (Hijmans,
2022). This was then repeated for the medium scale (1:5,000,000) and fine scale (1:1,000,000)
layers with each river raster overlayed with the previous coarser scale layer to ensure all rivers
were included as the scales decreased.
To approximate the exposure setting of each cell, data on the mean distance from shore of
surrounding cells was used to construct a proxy value of fetch. Using the *terra.focal* function
(Hijmans, 2022), the mean distance to shore of surrounding pixels was calculated in square
windows of width 10 km, 20 km, 50 km, 100 km, 175 km and 250 km. Due to extremely inhibitive
computational times when calculating these values at the two largest distances, the distance to
shore layer was first aggregated to a 400 m resolution before focal calculations of these
components, and then disaggregated back to 200 m to maintain uniformity. The maximum value
in each layer was then set to the relative window size, and all data in each layer normalised
between 0 and 1. The mean of all layers was then calculated which resulted in continuous
measure of relative exposure/shelteredness ranging from 0 (highly sheltered) to 1 (highly
exposed).

### 237    2.3.3 Satellite derived predictors

Using data from the Copernicus Marine Data Store, two layers were created approximating the
mass of suspended particulate matter in surface waters and the orbital velocity of waves at the
seafloor. Data on suspended particulate matter in surface waters across Canada from 2007 to
2019 was extracted in netCDF format from ACRI-ST (Sophia Antipolis, France) company's global
Bio-Geo-Chemical products at 4 km spatial resolution and a monthly temporal resolution
(Copernicus, 2022b). The climatological mean across this entire period was then calculated for



each cell and the netCDF converted to a raster for further processing. Due to the complex nature
of the Canadian coastline and the large dissimilarity in spatial resolution of the satellite data
product (4 km) and the layers created above (200 m), the satellite raster layer was allowed to
extrapolate by 1 cell in its native resolution by taking the mean value of neighbouring pixels. This
allowed better overlap of satellite layers with the study area maxima at the coastline but limited
over-extrapolation. The raster layer was then reprojected to the equal area CRS and resampled
onto the bathymetry layer using cubic-spline interpolation. Due to a lack of consistent SPM data
recorded in the northern Arctic Basin, this portion of the data layer was manually removed within
QGIS.
To calculate the estimated orbital velocity of waves at the seafloor, two satellite wave data
products were combined with the unified bathymetry layer as constructed above. Hourly data from
2007 to 2019 on the significant wave height ($H_s$; VHM0) in meters, and primary wave swell mean
period ($T_z$; VTM01_SW1) in seconds, were extracted from the 0.2° resolution Global Ocean Wave
Reanalysis (WAVERYS) produced by Mercator Océan International (Copernicus, 2022c) and the
3 km resolution Arctic Ocean Wave Hindcast produced by MET Norway (Copernicus, 2022a). All
data were processed as the SPM data layer (except for lack of removal of the Arctic basin data),
and converted to an estimate of orbital wave velocity at the seafloor ($U_{rms}$; measured in m s$^{-1}$)
using the following equation from Soulsby (2006);
$$U_{rms} = \left(\frac{H_s}{4}\right)\left(\frac{g}{d}\right)^{0.5} \exp\left\{-\left[\left(\frac{3.65}{T_z}\right)\left(\frac{d}{g}\right)^{0.5}\right]^{2.1}\right\} \qquad (1)$$
where $g$ is the acceleration due to gravity (9.806 m/s$^2$) and $d$ is the water depth (m), taken as the
unified bathymetry layer multiplied by -1, and all values less than 1 meter depth rounded up to
the nearest meter (as needed for the above calculation). The resultant Arctic orbital velocity data
layer was then bias corrected to the global orbital velocity data layer utilising the *qmap* package
with quantile mapping using a smoothing spline (Gudmundsson et al., 2012). Finally, the two data
layers were overlayed with the regional Arctic data taking priority over the global data where
available.

2.3.4 Ocean circulation model predictors
Data on the mean surface ice cover, seafloor salinity, temperature and current velocity was
collated from three different ocean circulation model products covering different regions of
Canada (Table 1). ANHA12 is a regional  configuration of the NEMO ocean and sea-ice model

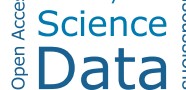

(Madec et al., 1998) created at the University of Alberta, covering the Arctic and northern
Hemisphere Atlantic at 5 day temporal resolution, a curvilinear 1/12th degree horizontal resolution
ranging from 1.93 km in the Arctic to 9.3 km at the equator, and 50 vertical levels (Hu et al., 2019).
The British Columbia continental margin (BCCM) circulation model created by Fisheries and
Oceans Canada (DFO) covers the entire Canadian Pacific coast and extends approximately
400 km offshore. It has a uniform horizontal resolution of 3 km, 42 vertical levels and a 3 day
temporal resolution (Peña et al., 2019; Masson and Fine, 2012). As the BCCM model has higher
uncertainty in nearshore and enclosed environments due to its relatively coarse resolution, data
was also extracted from the Salish Sea Cast ERDDAP data server. Similarly to the ANHA12
model, the Salish Sea Cast is a configuration of the NEMO circulation model developed by a
consortium of Canadian Universities and government agencies and extends from Juan de Fuca
Strait to Puget Sound to Johnstone Strait at 500 m horizontal resolution, 40 vertical layers and
hourly temporal resolution (Soontiens and Allen, 2017; Soontiens et al., 2016). For further details
on all these models, see relevant cited references. It should be noted that many of these ocean
circulation models contain high uncertainty in nearshore areas. However, they are expected to be
greatly improved when compared to global circulation model products which are frequently used
in this sort of predictive mapping work (e.g. Atwood et al., 2020; Lee et al., 2019; Assis et al.,
292 2018).

Three-dimensional data for salinity, temperature, u-velocity (eastward) and v-velocity (northward)
was extracted from each model and the climatological mean across all time points between 2007-
2019 was calculated. For each horizontal cell, the seafloor value was taken as the lowest vertical
cell within a given position. Individual model outputs were then converted to spatial point data
using the cell centroid positions and transformed to the unified equal area CRS. Point data was
then converted to rasters with the respective resolution of each model, and the mean value taken
if two points from the same model lay within a single raster cell as an artifact of reprojection. As
the ANHA12 model has a varying horizontal resolution, point data were rasterized using the
smallest resolution of the original model (1.6 km) and then interpolated using the *gstat* package
(Gräler et al., 2016) and a nearest neighbour interpolation method (including cells for land within
the original model grid to supress extrapolation). For all three models, mean current velocity was
then calculated as the root mean square of the u-velocity and v-velocity values in each cell.
Finally, as carried out for the satellite data layers, each raster was allowed to extrapolate by one
cell in its native resolution (or for the case of the ANHA12 model - its median resolution) and
resampled onto the 200 m bathymetry grid using cubic-spline interpolation. The three rasters were
then combined; the Salish Sea Cast data only being applied to cells that lay within the Salish Sea

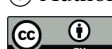



bioregion (as calculated in Section 2.3.2), the BCCM model outputs only being assigned to other
bioregions within the Pacific and ANHA12 used for all Atlantic and Arctic regions. Although this
means that different model products were used to measure the same predictor variable in different
regions, which can create biases, the bioregion predictor variable was included as a co-variate in
all models which included the ocean circulation variables, thus allowing for interactive effects and
accounting for differences in circulation model structures.
Predictor layers describing the mean concentration and thickness of sea ice for the same temporal
period across the Arctic and Atlantic were also derived from the ANHA12 model. Processing of
model data and spatial rasters was conducted as above, except a value of zero ice concentration
and thickness was applied to all cells across the British Columbia Pacific bioregions.

2.3.5 Global model predictors
Four additional predictor variables were derived from Bio-ORACLE version 2.2 - a global unified
marine environmental data-layers collation which gives climatological mean values at $1/12^{th}$
degree resolution, for 2000-2014 and a wide-range of environmental variables (Assis et al., 2018).
Although these datasets are expected to be of lower accuracy when compared to the regional
data used above, based on previous research there were some additional variables not available
from the regional circulation models which were considered potentially important for carbon
modelling (Diesing et al., 2021; Atwood et al., 2020). Three described the oceanographic
chemistry/biology – namely primary production and chlorophyll content of the surface water
column, and dissolved oxygen concentration at the seafloor. The fourth predictor was an
additional measure of current velocity (maximum current velocity), which was selected on top of
the previously derived mean values because current velocity has been identified as a particularly
strong predictor within previous seafloor sediment composition and carbon content predictive
mapping studies (Gregr et al., 2021; Diesing et al., 2021; Mitchell et al., 2019). Raster data were
downloaded from the Bio-ORACLE website and processed as the satellite data layers.

2.3.6 Final collation of predictor variables
The resulting 28 predictor variable raster layers were combined into a single raster stack and any
cells containing NA values removed, leaving only those cells which contained values across all
predictor layers. The remaining cells covered 92.3% of the subtidal zone of the Canadian EEZ





and delineated the maximum potential modelling area (Fig. 1). The final predictor variable layers
are shown in the Supplement.

**2.4 Sediment composition data**

Sediment composition point data were extracted from two sources. Firstly, all data were exported
from the NRCan Expedition Database on 11th November 2022. This data repository contains
information related to marine and coastal field surveys conducted by or on behalf of the Geological
Survey of Canada from the 1950s to present, which deployed sampling methods including piston
cores and grab samples. Data were also extracted from a recent synthesis of grain size
distribution measurements from the Canadian Pacific seafloor (1951-2017), compiled by
Geological Survey Of Canada and NRCan (Enkin, 2023). Although there are some duplications
between these two datasets, these are accounted for in the proceeding pre-processing steps. In
both sources, grain size data is reported as the percentage content of mud (sometimes separated
into silt and clay), sand and gravel within each sample. Due to modern developments in grain size
analyses (e.g. laser diffraction) older samples may have lower measurement accuracy; however,
due to the relatively coarse metric being used in this study (%mud/sand/gravel) and the
occurrence of a number of largescale geological surveys occurring during the 1960s, we chose
to retain data from 1960 onwards. Where sampling year was not recorded within the database,
the date was inferred from the expedition code or from expedition metadata. The sampling method
and depth of the sediment from which the sample/sub-sample originates are also predominantly
recorded within the database. Where sediment depth was absent, but the sampling method was
noted as "grab" or "other", the penetration depth was assumed to be 10 cm (a commonly assumed
penetration of standard sediment sampling devices such as Van Veen Grabs and Day Grabs).
Samples were only retained if they originated from within the top 30 cm of the sediment and had
associated geographic position information (latitude-longitude co-ordinates; lat-lon). Data were
further filtered by excluding those where the sum of mud, sand and gravel was greater than 102%
and lower than 98% - to allow for rounding errors but to exclude invalid data. Data were also
excluded if samples/sub-samples were not present from at least the top 1 cm to 5 cm below the
sediment surface within a given sampling event. After data filtering, the mean percentage of mud
was taken across replicates/sub-samples, leaving a single value for each sampling event. We
chose to concentrate on sediment mud content as this has previously been identified as the key
sediment composition component from a number of related carbon mapping studies (Smeaton et
al., 2021; Diesing et al., 2017, 2021; Pace et al., 2021; Wilson et al., 2018). Finally, mud content



data were projected onto the CRS of the predictor layers and only retained where overlap
occurred. This led to a final dataset of 19,730 samples (Fig. A1).

**2.5 Organic carbon content data**
2.5.1 Organic carbon data collation and extraction
Data on the percent organic carbon content within dried surface sediments (%OC) was collected
from three different structured searches. Firstly, a systematic literature review was conducted
through Web of Science and Scopus. Both searches were conducted on the 21st September 2022.
Within Web of Science, its "Core collection" was searched via the field "Topic", which examines
a paper's title, abstract, author, keywords and "keywords plus". Within Scopus, the search was
run via the field "Title-Abs-Key", which scans a paper's title, abstract and keywords. Within both
databases the same search string was used:
("organic carbon" OR "organic matter" OR "organic content" OR TOC OR TOM) AND (coast* OR
sea* OR ocean* OR estuar* OR marine OR gulf) AND (sediment* OR mud* OR sand* OR clay*
OR silt* OR gravel* OR seabed) AND Canad*
All articles identified from the searches were exported into a single Zotero library and duplicates
removed, leaving 1,581 results. Screening was conducted via a hierarchical process that first
assessed the title, then abstract and finally full text. At each stage an article was assessed against
the inclusion criteria described below, with those considered relevant or of unclear relevance
passing to the next level of assessment.
The inclusion criteria were defined as: 1) Study conducted on subtidal seabed sediments (those
concerning rock, shale or fauna were not included); 2) Physical samples collected using a seabed
sediment sampling device (e.g. cores or grabs - sediment-trap samples were not included); 3)
Samples from within the Canadian EEZ; 4) Studies concerning the chemical composition of the
sediment; 5) Organic carbon content (%) directly measured after separation of organic and
inorganic components (e.g. by acidification). After the title screening stage 242 articles remained,
followed by 123 remaining after abstract screening, and a final set of 49 articles left for data
extraction after review of the full text. Four additional primary literature papers were added based
on expert advice.
The second structured search was conducted on the Canadian Federal Science Libraries Network
- a repository which contains departmental publications, reports and data sets from seven



science-based Canadian government departments. The search was carried out on the 7th
November 2022 using the same search string as for the primary literature and querying all fields.
The search led to only 178 results and therefore each result was assessed individually against
the selection criteria first by their abstract and then by a full text assessment, leading to data
extraction from 15 reports. The third search was carried out on the 15th November 2022 using
GEOSCAN - the NRCan bibliographic database for scientific publications. As GEOSCAN does
not allow search strings containing "AND", the search was conducted on all fields using only the
terms: "organic carbon" OR "TOC" OR "OC"; leading to 655 search results. The metadata of all
entries was exported as a text file and further refined using a secondary manual search for the
remainder of the search terms listed above within Microsoft Excel. This led to a final set of 233
results, 178 which were excluded by screening of the title, and a further 51 excluded by abstract
or full text screening, leaving 4 reports for data extraction.
In total, these three structured searches of primary literature and government reports led to 72
individual entries for data extraction. As well as data on the %OC, metadata extracted included
the maximum depth of sample into the sediment (cm), geographic position (lat-lon), sample ID,
year of sampling (approximated as publication year where not clearly stated), sampling method
(e.g. multicorer, Van Veen grab) and water depth of sample site (where recorded). Data were
extracted from data tables or supplementary databases when available, otherwise the
PlotDigitizer online application was used to extract data from graphical products. Where possible
data were extracted on the %OC in different depth-layer sub-samples through a single core-
sample up to 50 cm, otherwise a single mean value was taken.
Additional to data collated through the structured searches, %OC data were also extracted from
PANGAEA - a global data repository for geographic earth-system data (PANGAEA®, 2022). A
data search across all topics was conducted on the 25th October 2022 using the same search
terms as for the structured search, except for removal of the term "*Canad\**".  The geographic
extent of the results was instead delineated using the spatial tool within PANGAEA which allows
results to be filtered by the geographic co-ordinates of a square/rectangular extent. Overall, this
led to a total of 1,489 potential datasets. All relevant data within these datasets were exported
using the Data Warehouse Download tool within Pangaea. Based on expert knowledge, two
additional PANGAEA datasets were added to the output from published global %OC data-
syntheses (Atwood et al., 2020; Seiter et al., 2004). Lastly, where the date of the sample was not
recorded, the sampling year was manually added by further exploring the metadata or cited
studies. To align the PANGAEA data with the systematic review data, PANGAEA data points





were excluded if: 1) they lacked data on %OC; 2) they lacked metadata on the depth of a sample
within the sediment; 3) if the sample originated from greater than 50 cm below the sediment
surface; or 4) metadata on the elevation/water depth indicated sampling above the subtidal.
Additionally, metadata within PANGAEA were coalesced where necessary (due to different
names being given to the same data type), and mean values of %OC taken if replicates were
measured within a single sub-sample.
All organic carbon data were converted into spatial point data, transformed to the unified equal
area CRS and masked by the predictor variable's maximum model area to leave only overlapping
data. Additionally, values were only retained from the sampling year 1959 and onwards. The extra
year was included when compared to the sediment composition data because there were some
widescale surveys undertaken across the Labrador Sea in 1959 which was lacking from any
additional %OC datasets. While this large temporal extent may add uncertainty in relation to the
quality and uniformity of the response data, similar extents have been used by previous global
mapping studies (Atwood et al., 2020; Lee et al., 2019; Seiter et al., 2004) and, 72% of the %OC
data within this study were sampled after 1980 and 55% after 2000. The larger temporal extent
also allows for the inclusion of a larger frequency and wider spatial extent of data, therefore
improving accuracy in spatial predictions. In total our %OC dataset contained 2,518 point-samples
(Fig. A2) and 3,308 sub-samples across different depth layers within cores.

2.5.2 Organic carbon data processing
Due to commonly adopted uneven sampling distributions within single core samples (i.e. more
sub-samples towards the top of the core), where sub-sample data were present on the %OC in
different depth-layers these were converted into weighted cumulative means assuming linear
distribution between sub-samples. Additionally, there was large variation in the maximum
sediment depth of point-samples, ranging from %OC measures from only the top 1 cm of
sediment, to values up to the chosen data extraction limit of 50 cm deep. We chose to standardise
all samples to 30 cm depth as only 6% of the point-samples covered sediment depths below this
layer and because 30 cm is a commonly suggested carbon stock accounting depth for terrestrial
soil and marine sediment habitats in both carbon accrediting methodologies and greenhouse gas
inventories (VERRA, 2020; IPPC, 2019).
To estimate the cumulative mean of %OC at 30 cm for all individual point-samples, we created a
transfer function using a generalised additive mixed model (GAMM) smoothing spline. It is



generally expected that the %OC in marine sediments decreases with depth within the seafloor
(Middelburg, 2018); we used the collated data above to approximate a mean decay function for
this study. Firstly, only those data that contained at least five sub-sampled depth layers were
retained for modelling as fitting distributions to those with fewer points would likely be invalid. This
left 183 unique samples with 2,640 weighted cumulative mean sub-samples for model
construction. Cumulative mean %OC data were arcsin transformed (arcsin{$\sqrt{}$ [%OC/100]}; a
commonly adopted transformation for percentage data), and a simple GAMM model applied with
sub-sample sediment depth as the fixed factor modelled with a cubic regression spline and
sample ID as the random factor. The GAMM model was fitted using the *mgcv* package; a scaled-
t distribution family was used for heavy tailed Gaussian-like data, the number of basis dimensions
was set to 20 and smoothing parameter estimation was conducted by Restricted Maximum
Likelihood (REML) (Wood et al., 2016). Model validation was carried out using visual assessment
of diagnostic plots of residuals, as well as observed vs fitted values. Significance of the sampling
depth smoothing spline was assessed by an analysis of variance (ANOVA) with a chi-squared
test comparing the full GAMM model to a null GAMM model containing only the random factor
and the intercept (see Appendix B for results). The difference between estimated deviance
explained in the full and null models was also used to approximate the variance explained by the
fixed and random factors. To create a transfer function, the cumulative %OC was predicted from
the mean fixed effects of the GAMM model at sediment depths from 0 – 30 cm at 0.1 cm intervals.
The predictions were then back-transformed to percentage data and the cumulative mean %OC
at each depth was converted to an inverse proportion of the mean at 30 cm. Overall, this gave an
estimated proportional conversion factor from the cumulative mean at any given depth to an
expected cumulative mean at 30 cm (Appendix B).
All point-sample data from PANGAEA and the systematic review were combined, corrected to
weighted cumulative means where sub-samples were present, checked for duplication, and
unified to a mean %OC value of the top 30 cm of sediment using the above transfer function. One
outlier was removed from the dataset as it was reported to have a carbon content twice that of
any other sample within the dataset. Finally, for further analyses %OC data were arcsin
transformed due to a highly right skewed distribution and its application within similar modelling
exercises (Smeaton et al., 2021; Diesing et al., 2017).




**2.6 Final model domain selection**

After visual assessment of the coverage of both the sediment composition and %OC data, the final model domain was limited to a water depth of 2,500 meters. This depth limit (as delineated by the bathymetry predictor layer) encompassed 99.95% of sediment composition point data (Fig. A1) and 99.3% of %OC data (Fig. A2). The predictor layer raster stack was filtered with all cells deeper than 2,500 meters excluded from the model domain. This final model domain covers 4,489,235 km$^2$ which is 78.4% of the EEZ or 92.6% of the seafloor area above 2,500 m (Fig. 1).

**2.7 Sediment mass accumulation rate data**

From preliminary exploratory research it was determined that there would be insufficient data on organic carbon accumulation rates, or sediment mass accumulation rates, to undertake a Canada-specific data synthesis. We therefore chose to downscale a recent global spatial predictive map of seafloor sediment mass accumulation rates (Restreppo et al., 2021). To approximate a sample of values across the model domain in this study the global mass accumulation rate data (MAR; $\log_{10}\{g\ cm^{-2}\ yr^{-1}\}$) netCDF was converted to a raster and masked by the coverage of the model domain. The raster layer was then converted to spatial point data by the location of cell centroids, and a stratified-random sample of 10% of the data was taken. Data was stratified by assigning the x-coordinate, y-coordinate and mass accumulation rate values to decile bins; and a random sample of 10% of values taken within each unique combination of the three-way binning. This resulted in 12,660 point estimates of MAR across the model domain, which were then reprojected to the unified equal area CRS for further analyses (Fig. A3).

**2.8 Random forest modelling**

For predictive mapping we adopted random forest machine learning techniques due to their flexibility regarding violations of traditional statistical assumptions, ability to handle a range of data types and predictor variables and elucidate both drivers of model response and predictions of uncertainty, as well as their successful application in previous similar modelling tasks (Diesing et al., 2017, 2021; Pace et al., 2021; Atwood et al., 2020; Wilson et al., 2018). Contemporary research in spatial machine learning techniques have highlighted that robust spatially-explicit cross-validation (CV) strategies and predictor variable selection processes are essential to




calculate valid performance metrics, limit overfitting and construct reliable spatial predictions
(Zhang et al., 2023; Ludwig et al., 2023; Meyer and Pebesma, 2022; Meyer et al., 2019). We
discuss the incorporation of these processes into our modelling framework below.
Three response variables (mud content, organic carbon content (%OC) and MAR) were modelled
using the following framework. Firstly, each response variable was overlain onto the predictor
variable grid and the mean values were taken if more than one data-point fell within a single raster
cell. All predictor variable data were then extracted for each response dataset; however, the three
biological/biochemical predictor variables (primary production, chlorophyll concentration and
dissolved oxygen) were only used within the %OC model as they are not expected to drive
variation in physical sediment properties (Restreppo et al., 2021; Gregr et al., 2021; Graw et al.,
2021; Mitchell et al., 2019).
For each response variable, the *spatialsample* package (Silge and Mahoney, 2023) was used to
construct a variety of spatial CV data-fold structures (splitting the data into different analysis and
assessment sets) and the validity of each structure was visually assessed using the
*CAST.plot_geodist* function (Meyer et al., 2023). This function creates density plots of nearest
neighbour distances in multivariate predictor space between all response data as well as between
response data and a random sample of prediction locations, and between analysis and
assessment data within CV folds (see Appendix C). The suitability of a given CV structure to be
representative of estimating map accuracy can be determined by visually assessing the density
plots and finding the analysis-to-assessment CV-distance curve being closely aligned to the
sample-to-prediction density curve (see Appendix C; Ludwig et al., 2023; Meyer and Pebesma,
2022). Contrastingly, if the sample-to-sample distance curve closely overlays the sample-to-
prediction curve, this indicates that traditional random cross-validation strategies are likely to be
appropriate (see Appendix C; Ludwig et al., 2023). To approximate sample-to-prediction
distances, the sample size number within *plot_geodist* was set to select 5,000 random samples
across the model domain. Further, as the spatial distribution of data is a key consideration to
ensure robust cross-validation (Ludwig et al., 2023; Meyer and Pebesma, 2022), the x- and y-
coordinates of each data point were also included as predictor variables in the *plot_geodist*
calculations.
For the mud content data, a spatial kmeans clustering CV structure was chosen as the response
data had good coverage of the model domain, contained a large number of data points, and
showed relatively strong spatial clustering (Fig. A1). A range of options in the number of kmeans
clusters were tested, with 35 being determined as the optimal number and each cluster being



assigned to its own CV fold (Fig. C1). Through visual assessment of the density plots, it was
identified that the kmeans CV structure was somewhat mis-aligned from response-to-prediction
distances, with the CV distances being overly conservative at including near-distance
comparisons (Fig. C1). We therefore used a partially repeated CV strategy, with a small number
of randomly selected data-points added to the assessment set in each kmeans spatial-CV fold
(1% of mud content data randomly sampled at each fold without replacement) (Fig. C2). As the
%OC response dataset was relatively small and spatially dispersed (Fig. A2), we used a spatial
block CV strategy in place of the kmeans clustering to avoid clusters containing highly spatially
dispersed data. We chose to use hexagonal shaped blocks, random assignment of blocks to folds,
and the same number of CV folds as for the mud content data (v = 35) - both to maintain uniformity
and because varying the fold-number did not significantly influence the density plots. Instead, the
diameter of the spatial blocks was altered, and an optimal block size of 100 km identified using
the *plot_geodist* function (Fig. C3). For both response variables, following identification of an
appropriate CV structure, a single fold was assigned as testing data, with all other data retained
for model fitting. Following the training-testing split, the spatial CV folds were reconstructed on
the training data to ensure an absence of duplication. For the MAR data, the density plots
indicated that traditional random cross-validation would be a valid approach (Fig. C4), which was
expected as the response data were a stratified-random sample across the model domain (Fig.
A3). The random CV folds were stratified by the MAR response value to ensure a relatively even
distribution across CV folds. A 10% stratified-random sample was first assigned as the test-set
and random CV folds assigned to the remaining training data.
Three random forest models were constructed (mud content, OC content and MAR), each
following the same modelling protocol. Firstly, the *CAST.ffs* function (Meyer et al., 2023) was used
to run a spatially-explicit forward predictor variable selection processes. The function fits a model
with all combinations of two-way predictors, selects the best model based on a given metric, and
then increases the number of predictors by one, testing all remaining variables. This iteratively
continues with the process stopping if none of the tested variables increases the performance
when compared to the best previous model with "n-1" predictors. The function also allows models
to be fit separately across all individual CV folds, therefore incorporating appropriate spatial
considerations into the feature selection process. Due to the large number of variables within this
study, and the relatively large datasets, this process was very computationally expensive. We
therefore chose to adapt the function to initiate forward variable selection after *a priori*
identification of the first two predictor variables. These variables were identified by constructing a
basic random forest model with all training data and predictor variables, and the hyperparameters



*mtry* (the number of variables to randomly sample as candidates at each split), *min_n* (the number
of observations needed to keep splitting nodes) and *trees* (the number of random forest trees to
construct and take mean predictions across) set to 2, 5 and 1,000 respectively. Variable
importance was estimated using permutation, and the two predictor variables with largest
importance selected. The *ffs* function was then run starting with the two pre-selected variables
(see Fig. 3, 6 & 9) and performance of each iteration assessed on the root mean squared error
(RMSE) of predictions across all CV folds.
Following variable selection, hyperparameter tuning was conducted on the *mtry* and *min_n*
hyperparameters, with the number of trees set to 1,000. The tuning process fitted individual
models across all CV folds, each with 11 combinations of hyperparameters which were selected
using a semi-random Latin hypercube grid. The performance of each hyperparameter
combination was assessed based on the RMSE of predictions across all CV folds. After selection
of the best performing hyperparameter combination, a last model fit was conducted on the entire
training set and evaluated on the test set, with the absence of overfitting determined by the RMSE
and $R^2$ of the last-fit model falling within the range of those found across CV folds. Overall model
performance metrics (RMSE and $R^2$) were then calculated using the predictions across all CV
folds with optimal hyperparameters and the last-fit; while predictor variable importance was
calculated by fitting an additional model across all training data using optimal tuning parameters
and the importance calculated through permutation. Accumulated local effects (ALE) plots were
produced for the six predictor variables with highest importance in each model using the *iml*
package (Molnar et al., 2018) to give a visual representation of the average effect of predictors
on model prediction outcomes. Finally, mean model predictions were calculated across the entire
model domain using the last-fit model and the predictor variable raster stack, and cell-specific
estimation of uncertainty was calculated using standard error on out-of-bag predictions using
infinitesimal jack-knife for bagging (Roy and Larocque, 2020; Wager et al., 2014). Due to
computational restraints when calculating predictions across the entire model domain (which
contains 112,230,871 cells), data were split into 150 random samples (without replacement) and
both prediction and standard error estimates made serially on each split. All predictions were then
merged to create a raster layer covering the entire model domain.
A cell-specific approximation of the upper and lower bounds of the 95% confidence interval (CI)
was calculated by adding/subtracting the cell-specific standard error estimates, each multiplied
by 1.96, from the mean predictions and then back transformed where needed (Kuhn and
Wickham, 2020; Wager et al., 2014). After calculation, CI values were corrected where necessary





- being bounded by 0, and where applicable also bounded by 100. The resulting three raster
layers from the mud content model were also used as available additional predictor variables
when constructing the random forest models for %OC and MAR as outlined above (Fig. 2).

**2.9 Estimating sediment dry bulk density**
To estimate the dry bulk density of the sediment across the model domain ($\rho D$ – the mass of dried
sediment per unit volume within the seafloor; g cm$^{-3}$) the outputs from the random forest models
for mud and organic carbon content were combined with a variety of published transfer functions
and global modelled products (Fig. 2). Three of the transfer functions calculate the porosity of the
sediment ($\Phi$; the proportion of sediment volume which is water) based on the predicted mud
content using the following equations, respectively from Jenkins (2005), Diesing et al. (2017) and
Pace et al. (2021):
$\Phi = 0.3805 . mud + 0.42071$                                              (2)
$\Phi = 0.4013 . mud + 0.4265$                                              (3)
$\Phi = 10^{\{0.138 . \log_{10}(mud) - 0.486\}}$                                  (4)
In all cases *mud* is the predicted values across the model domain as calculated above expressed
as a decimal proportion. For Equation 4 mud content was rounded up to the nearest 0.01 as lower
values give unrealistic porosity estimates. All sediment porosity estimates were then converted to
an estimate of dry bulk density using the following equation:
$\rho D = \rho S (1 - \Phi)$                                                       (5)
where $\rho S$ is the grain density of seabed sediments in g cm$^{-3}$, which was set at the frequently used
constant approximation of 2.65 (Diesing et al., 2017, 2021; e.g. Pace et al., 2021; Lee et al., 2019;
Wilson et al., 2018; Kuzyk et al., 2017). Although this standard approximation of grain density is
not ideal, the variation under different environmental settings is generally found to be small when
compared to differences in %OC and porosity, therefore the values of grain density are not
expected to strongly drive variation in organic carbon density (Atwood et al., 2020; Lee et al.,
2019; Middelburg, 2019; Martin et al., 2015; Berner, 1982). A forth transfer function from Atwood
et al. (2020) calculates an estimate of dry bulk density directly from %OC using the following
equation:
$\rho D = 0.861 . \%OC^{-0.3999}$                                            (6)



For this equation, carbon content as predicted above was rounded up to the nearest 0.1% as
lower values give unrealistic dry bulk density estimates. For each of the four transfer functions
(Equations 2,3,4 and 6) the value was calculated using the mean prediction as well as the upper
and lower confidence interval bounds of mud content and %OC respectively, resulting in three
raster layers from each function.
Two further estimates of dry bulk density were calculated using products from global predictive
models, both at 5 arc min spatial resolutions. Martin et al. (2015) created a predictive map of
seabed sediment porosity, while Graw et al. (2021) estimate sediment wet bulk density ($\rho W$)
across the global seafloor. Both raster layers were processed as the satellite predictor layers to
align with the model domain. The resulting porosity raster layer was converted to dry bulk density
using Equation 5, while the wet bulk density layer was initially converted to porosity using the
equation:
$$\Phi = \frac{\rho W - \rho S}{\rho SW - \rho S} \qquad (7)$$
where $\rho SW$ is the density of seawater estimated as 1.024 g/cm$^3$. In total this led to 14 dry bulk
density estimates across the model domain. A final mean value and standard error was calculated
for each cell, and the upper and lower 95% confidence interval bounds calculated using the
standard error as above.

**2.10 Estimating organic carbon standing stock and accumulation rates**

The organic carbon density (g cm$^{-3}$) is calculated by multiplying the %OC (expressed as a decimal
proportion) by the sediment dry bulk density; while organic carbon accumulation rates (g cm$^{-2}$ yr$^{-1}$)
are calculated by multiplying MAR by %OC (Fig. 2). For the final calculations of both density
and accumulation the respective means, upper and lower CI bounds were multiplied together to
incorporate uncertainty from both components. To create more meaningful response values
organic carbon density was converted to kg m$^{-3}$ (multiplied by 1000) and organic carbon
accumulation to g m$^{-2}$ y$^{-1}$ (multiplied by 10,000). Finally, the organic carbon stock in each mapped
cell can be calculated by multiplying the organic carbon density by the reference sediment depth
of this study (0.3 m) and the cell area (40,000 m$^2$) and converted to metric tonnes (divided by
1000). The total accumulation per cell per year can be calculated by multiplying the organic carbon
accumulation rate by the cell area. Overall, this allows estimates to be calculated for the total
values of organic carbon stock and accumulation across different parts of model domain.






**2.11 Rock substrate distribution case studies**

The method followed in this study is similar to that used for many similar predictive mapping exercises in that it uses data only from sediment grab and core samples to build predictive maps across the model domain (Restreppo et al., 2021; Graw et al., 2021; Diesing et al., 2017, 2021; LaRowe et al., 2020a; Atwood et al., 2020; Lee et al., 2019; Mitchell et al., 2019; Wilson et al., 2018; Stephens and Diesing, 2015). One major limitation with this modelling approach is that areas of bedrock, which would have zero values for all sediment response variables, will not be recorded in these datasets. Therefore, the under representation of zero values in the response data could lead to an overestimate of organic carbon standing stocks and accumulation rates as zero values are unlikely to be predicted from model outputs.

In the context of this study, information regarding the distribution of bedrock is lacking for many regions. We therefore use two regional case studies from the Pacific British Columbian EEZ and the Atlantic Scotian shelf and slope where recent publications have made estimated maps on the distribution of rock substrates (Philibert et al., 2022; Gregr et al., 2021). Each of these products was overlaid onto the final spatial predictions of sediment carbon densities and accumulation rates and all cells set to zero where rock substrates were predicted. The proportional effect on the mean, upper and lower confidence interval bounds of estimated carbon stock and accumulation rates was then calculated in each bioregion.

**3. Results**

**3.1 Mud content predictive mapping**

Of the 25 predictor variables available for mud content random forest modelling, 13 were selected in the optimal model (Fig. 3). Mean orbital velocity of waves at the seafloor and the mass of suspended particulate matter at the surface were the variables with highest importance (Fig. 3). Other variables with relatively high importance for predicting mud content included the exposure setting, ice thickness, distance to rivers, bathymetry, and benthic position indices (Fig. 3). Higher mud content was generally predicted in areas of low wave velocity, low exposure and close to but not directly adjacent to river mouths; with the effect of SPM and ice thickness less distinct, likely due to more complex interactive effects (Fig. 4).

723

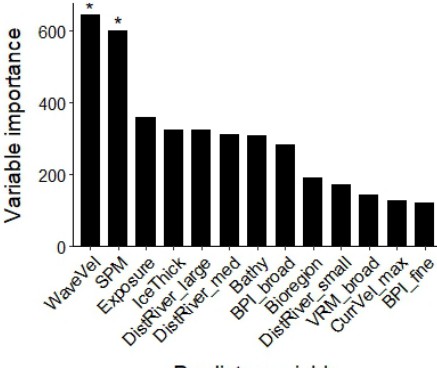

724

**Figure 3. Predictor variable importance from random forest models of mud content in marine subtidal sediments.** The y-axis is a unitless relative variable importance score for each model. Asterisks indicate the *a priori* variable selection. WaveVel = Orbital wave velocity at the seafloor, SPM = Suspended particulate matter within the water column, BPI = Benthic position index, DistRiver = Distance to nearest river, IceThick = Sea ice thickness, Bathy = Bathymetry, VRM = Vector ruggedness measure, CurrVel = Current velocity at the seafloor.

730

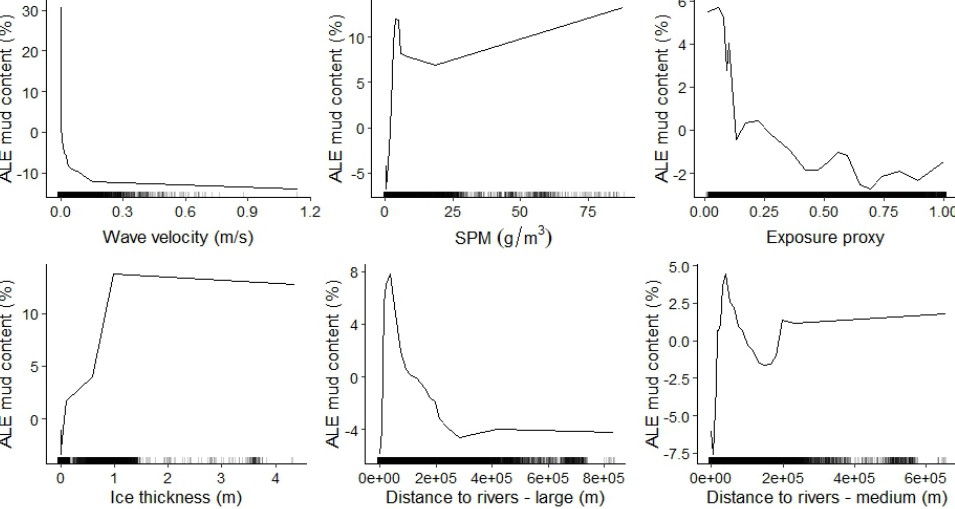

731

**Figure 4. Accumulated local effects (ALE) plots for the six predictor variables with highest importance in the mud content random forest model.** ALE gives a visual representation of the average effect of the predictor variable on the response but does not indicate the influence of multi-way interactions which are inherent in random forest models. Rug plots indicate the distribution of each variable within the training dataset. SPM = suspended particulate matter.



737

Areas with sediments dominated by mud (>75%) were predicted across the basins of many of the
Pacific fjords, inlets and estuaries, and within the southern Salish Sea (Fig. 5). In the Arctic, mud
dominated areas included large parts of the Canadian western Arctic as well as Hudson Bay. In
the Atlantic, the Laurentian channel and central Scotian Shelf contained particularly high mud
fractions (Fig. 5). Across the model domain, sediment in deeper areas on the continental slope
was also highly dominated by mud (Fig. 5) Using robust spatial cross validation, the model was
estimated to have an RMSE of 24.4% and $R^2$ of 0.60. The cell specific upper and lower 95% CI
bounds are shown in Figure D1. On average the upper CI bounds were 28% higher than the mean
and the lower CI bounds 20% less.

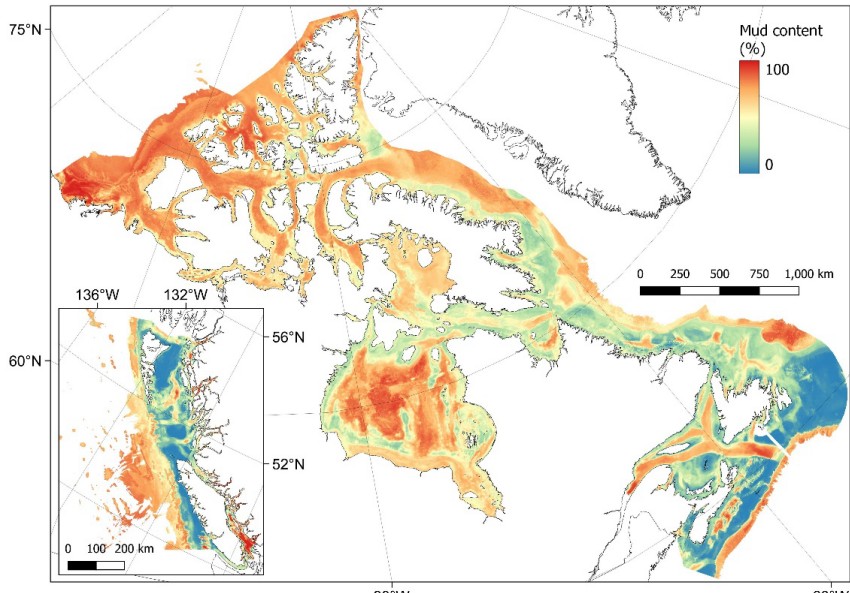

747

**Figure 5. Predictive mapping of mud content (%) in subtidal marine sediments across the Canadian continental margin.** The main plot shows the Arctic and Atlantic regions with the Pacific region inset. The 95% confidence interval bounds around the predicted means are shown in Figure D1. Labels indicating the locations of different areas mentioned within the text are shown in Figure A4. Country outlines from World Bank Official Boundaries, available at https://datacatalog.worldbank.org/search/dataset/0038272.



### 3.2 Organic carbon content predictive mapping

Eleven predictor variables were selected in the optimal organic carbon content (%OC) model (Fig. 6). The variables with highest importance in predicting %OC were the mud content layers constructed above (specifically the mean and lower CI bound), with all other predictors having less than half the relative importance of the mean mud predictions (Fig. 6). On average organic carbon content increased with predicted mud content and was generally higher in areas with low SPM concentrations, low exposure settings, close to but not directly adjacent to rivers, and at high water temperatures (Fig. 7).

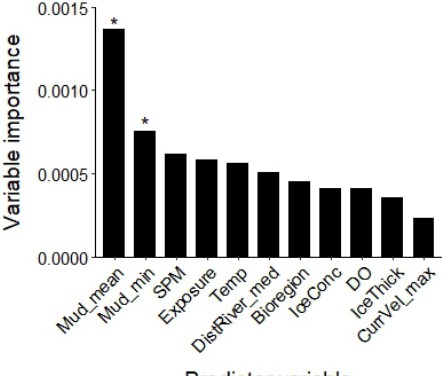

**Figure 6. Predictor variable importance from random forest models for the organic carbon content in marine subtidal sediments.** The y-axis is a unitless relative variable importance score. Asterisks indicate the *a priori* variable selection. Mud_min = Lower bound of 95% CI for mud content, SPM = Suspended particulate matter within the water column, Temp = Temperature, DistRiver = Distance to nearest river, IceConc = Sea ice concentration, DO = Dissolved oxygen at the seafllor, IceThick = Sea ice thickness, CurrVel = Current velocity at the seafloor.

771

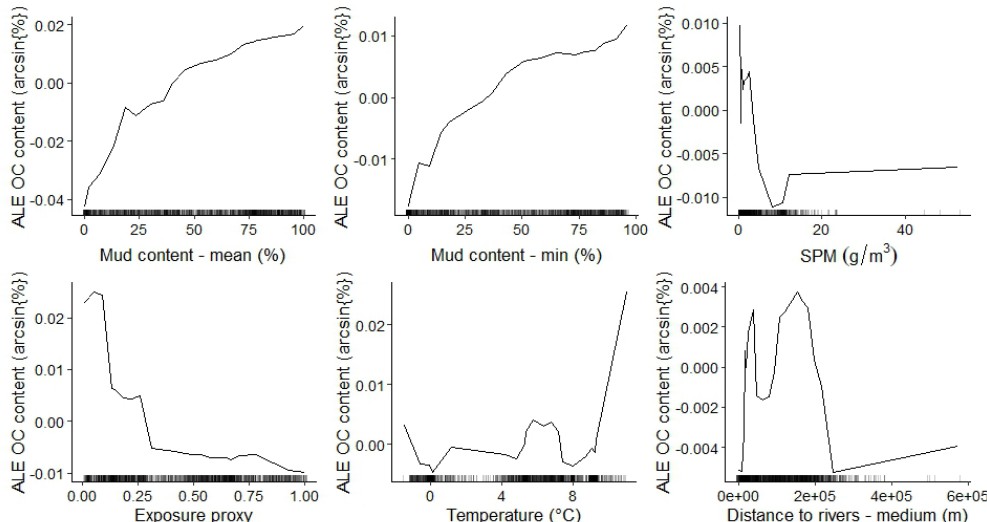

772

**Figure 7. Accumulated local effects (ALE) plots for the six predictor variables with highest importance in the organic carbon (OC) content random forest model.** ALE gives a visual representation of the average effect of the predictor variable on the response but does not indicate the influence of multi-way interactions which are inherent in random forest models. Rug plots indicate the distribution of each variable within the training dataset. SPM = suspended particulate matter.

The predictions of %OC ranged from $3\times10^{-5}$ to 5.6% with an overall mean of 0.8 ± 0.3% (± SD). Areas with highest predicted %OC (>3%) were restricted to parts of the Pacific west coast fjords and channels, and in small parts of the inlets and bays on the east coast of Nova Scotia and around Passamaquoddy Bay in the Bay of Fundy (Fig. 8). High concentrations (i.e. >1%) were more widespread across these areas as well as covering much of the Beaufort Sea, western Baffin Bay and Foxe Basin in the Arctic, southern and central Hudson Bay, the Laurentian channel, coastal north Newfoundland and the central Scotian shelf in the Atlantic, as well as across the Salish sea and deeper areas to the south of the British Colombian continental margin (Fig. 8). Lowest %OC was predicted across shallower parts of the central Pacific shelf and near coast areas west of Vancouver Island (Fig. 8). Cross validation estimated an $R^2$ for the model of 0.58 and an RMSE of 0.09 arcsin{%OC}. Cell specific upper and lower 95% CI bounds are shown in Figure D2. On average the upper CI bounds were 42% higher than the mean prediction, and the lower CI bounds 33% less than the mean prediction.

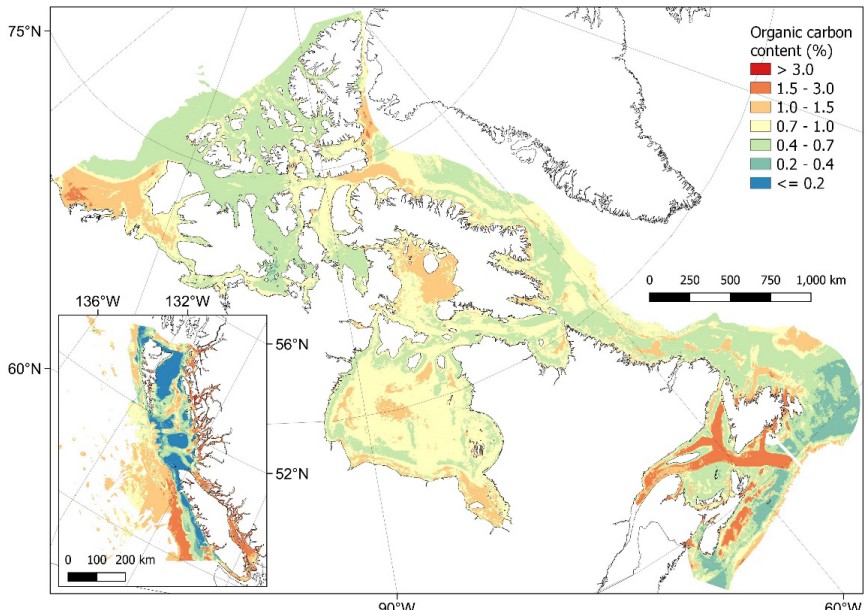

791

**Figure 8. Predictive mapping of organic carbon content (%) in subtidal marine sediments across the Canadian**
**continental margin.** The main plot shows the Arctic and Atlantic regions with the Pacific region inset. The continuous
variable is shown displayed in discrete colour bands to improve visualisation of highly right skewed data. The 95%
confidence interval bounds around the predicted means are shown in Figure D2. Labels indicating the locations of
different areas mentioned within the text are shown in Figure A4. Country outlines from World Bank Official Boundaries,
available at https://datacatalog.worldbank.org/search/dataset/0038272.

## 3.3 Sediment mass accumulation rate predictive mapping

The optimal model for mass accumulation rate (MAR) of seabed sediments contained 10
variables (Fig. 9). On average, MAR was negatively associated with increasing ice thickness, ice
concentration, salinity and distance from rivers, and was particularly high in Eastern bioregions
(Fig. 10). The predictions of MAR ranged from $4\times10^{-4}$ to 0.35 g cm$^{-2}$ yr$^{-1}$ with an overall mean of
$0.01 \pm 0.03$ g cm$^{-2}$ yr$^{-1}$ ($\pm$ SD). Areas with highest MAR (>0.1 g cm$^{-2}$ yr$^{-1}$) were predicted on the
east coast around inshore areas of the Gulf of St Lawrence and Bay of Fundy (Fig. 11). Other
areas with higher than average MAR were predicted across Canadian inshore areas particularly
in the southern Arctic, Hudson Bay, Foxe Basin, Salish Sea and northeast British Colombia Pacific
shelf (Fig. 11). The optimal model had an estimated $R^2$ of 0.89 and RMSE of 0.206 log$_{10}${g cm$^{-2}$
yr$^{-1}$}. Cell specific upper and lower 95% CI bounds are shown in Figure D3. On average the upper
CI bounds were 33% higher than the mean prediction, and the lower CI bounds 20% less than
their means.



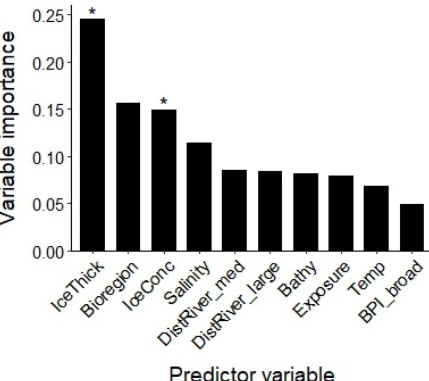

811

**Figure 9. Predictor variable importance from random forest models for the mass accumulation rate of subtidal sediments.** The y-axis is a unitless relative variable importance score**.** Asterisks indicate the *a priori* variable selection. IceThick = Sea ice thickness, IceConc = Sea ice concentration, DistRiver = Distance to nearest river, Bathy = Bathymetry, Temp = Temperature, BPI = Benthic position index.

816

817

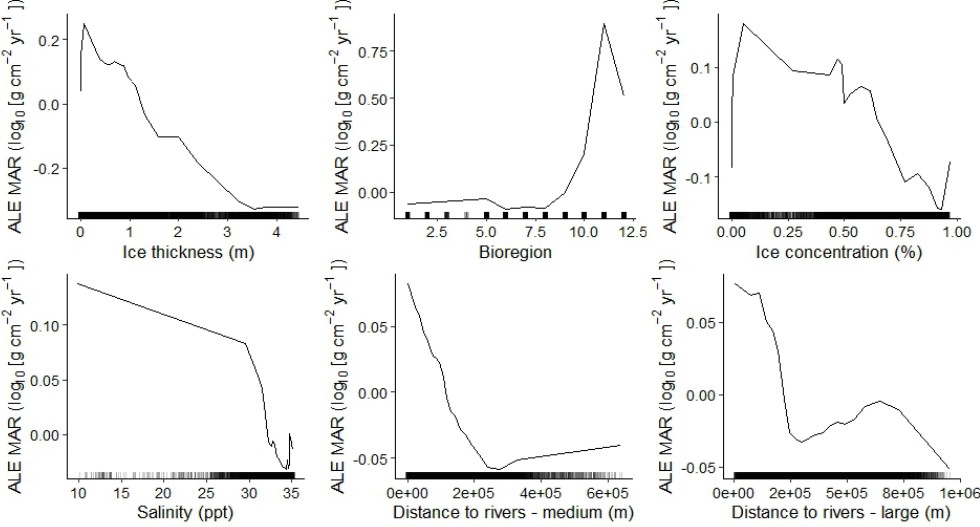

818

**Figure 10. Accumulated local effects (ALE) plots for the six predictor variables with highest importance in the sediment mass accumulation rate (MAR) random forest model.** ALE gives a visual representation of the average effect of the predictor variable on the response but does not indicate the influence of multi-way interactions which are inherent in random forest models. Rug plots indicate the distribution of each variable within the training dataset.

823
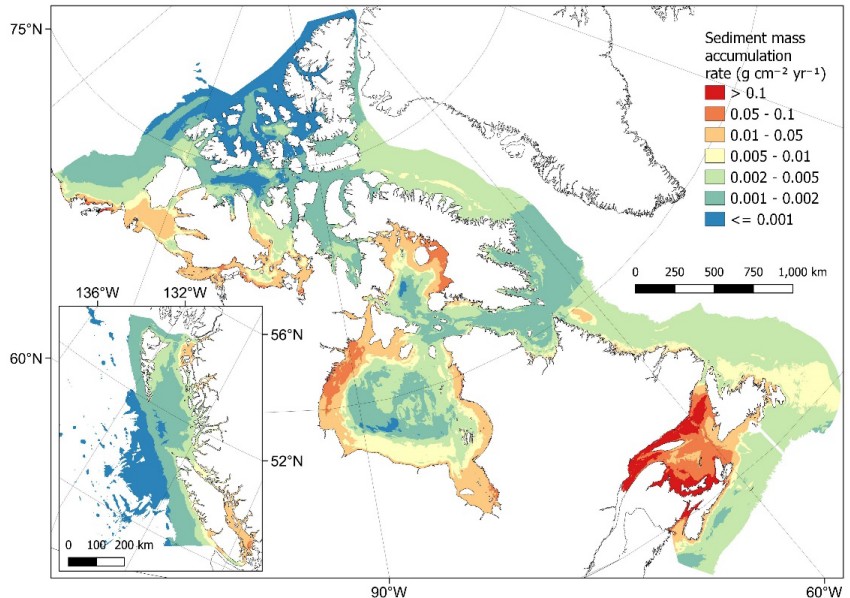

824

**Figure 11. Predictive mapping of sediment mass accumulation rate (g cm$^{-2}$ yr$^{-1}$) across the Canadian continental margin.** The main plot shows the Arctic and Atlantic regions with the Pacific region inset. The continuous variable is shown displayed in discrete colour bands to improve visualisation of highly right skewed data. The 95% confidence interval bounds around the predicted means are shown in Figure D3. Labels indicating the locations of different areas mentioned within the text are shown in Figure A4. Country outlines from World Bank Official Boundaries, available at https://datacatalog.worldbank.org/search/dataset/0038272.

831

**3.4 Dry bulk density estimation**

The dry bulk density of sediments was estimated using a variety of transfer functions and global predictions (Fig. 2). Estimated values ranged from 0.67 – 1.62 g cm$^{-3}$ with a mean of 1.02 ± 0.16 g cm$^{-3}$ (± SD). As many of the transfer functions are dependent on the predicted mud content, the spatial distribution of dry bulk density values was very similar to the mud content values predicted above (Fig. 5), i.e. lowest dry bulk density was estimated in mud dominated areas (Fig. 12). Cell specific upper and lower 95% CI bounds are shown in Figure D4. On average CI bounds were 8.5% either side of their means.

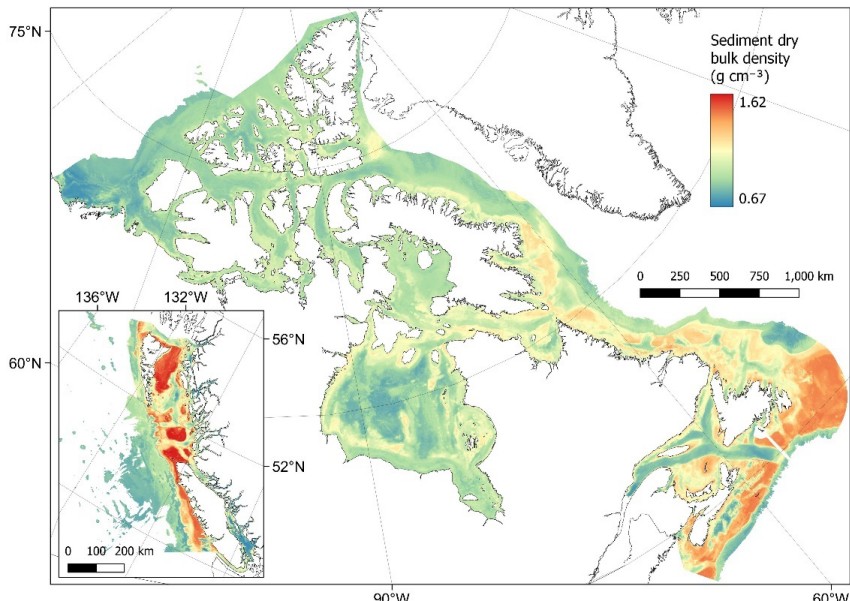

840

**Figure 12. Estimates of sediment dry bulk density (g cm⁻³) across the Canadian continental margin.** The main
plot shows the Arctic and Atlantic regions with the Pacific region inset. The 95% confidence interval bounds around the
predicted means are shown in Figure D4. Labels indicating the locations of different areas mentioned within the text
are shown in Figure A4. Country outlines from World Bank Official Boundaries, available at
https://datacatalog.worldbank.org/search/dataset/0038272.

## 3.5 Estimated organic carbon density and standing stock

From combining predictions of dry bulk density and organic carbon content, organic carbon
density could be estimated across the Canadian continental margin (Fig. 2). Estimated values
ranged from $5 \times 10^{-4}$ to 50.0 kg m⁻³ with a mean of 7.9 ± 2.5 kg m⁻³ (± SD). Spatial patterns in
organic carbon density (Fig. 13) were similar to those found for organic carbon content (Fig. 8).
Areas with highest carbon density (> 25 kg m⁻³) were restricted to small areas within nearshore
zones, including inlets and fjords of British Columbia, as well as enclosed nearshore areas of the
Atlantic East Coast (Fig. 13). High carbon densities (> 15 kg m⁻³) where predicted to occur across
wide parts of these areas as well as further offshore in parts of the Laurentian channel and central
Scotian Shelf, and at the edge of the continental slope off the West of Vancouver Island (Fig. 13).
In the Arctic, areas with relatively high carbon (>10 kg m⁻³) were predicted across many nearshore
areas, as well as across large parts of the Beaufort Shelf, Foxe Basin, James Bay and the Kane
Basin (Fig. 13). Cell specific upper and lower 95% CI bounds are shown in Figure D5. On average

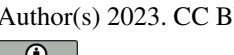

the upper CI bounds were 54% higher than the mean prediction, and the lower CI bounds 39%
less than their means.
Using a standardised sediment depth of 30 cm, the total standing stock of organic carbon in
surficial sediments across the model domain is estimated at 10.7 Gt with a 95% confidence
interval of 6.6 – 16.0 Gt. Between bioregions, total stock was predominantly related to the total
areal extent, for example Hudson Bay having the largest carbon stock and largest area (Table 2).
The Strait of Georgia and Southern Shelf bioregions of the Pacific had the lowest total standing
stocks due their small extent, however per unit area, these regions contained the highest organic
carbon stocks.

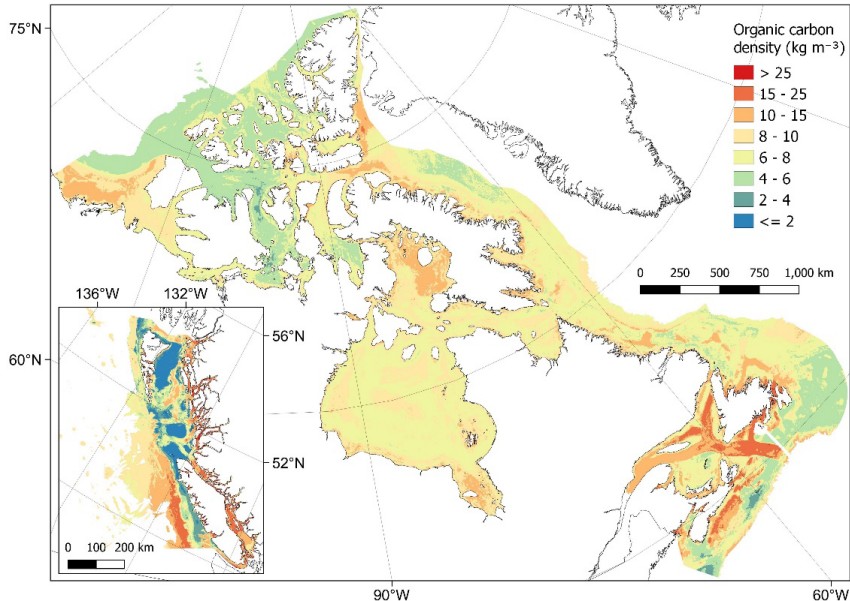


**Figure 13. Estimates of organic carbon density (kg m$^{-3}$) across the Canadian continental margin.** The main plot
shows the Arctic and Atlantic regions with the Pacific region inset. The continuous variable is shown displayed in
discrete colour bands to improve visualisation of highly right skewed data. The 95% confidence interval bounds around
the predicted means are shown in Figure D5. Labels indicating the locations of different areas mentioned within the
text are shown in Figure A4. Country outlines from World Bank Official Boundaries, available at
https://datacatalog.worldbank.org/search/dataset/0038272.





**Table 2. Summary of estimated mean total organic carbon stocks and accumulation rates in surficial seabed**
**sediments of different bioregions across the Canadian continental margin.** Organic carbon standing stocks are
estimated for the top 30 cm of seabed sediments. For delineation of the different bioregions see Supplement.

| Bioregion | Model domain extent (km²) | OC stock (Gt) | Stock per unit area (kt km²) | OC accumulation (Mt y⁻¹) | Accumulation per unit area (t km² y⁻¹) |
|---|---|---|---|---|---|
| 1. Offshore Pacific | 53,598 | 0.14 | 2.67 | <0.01 | 0.09 |
| 2. Northern Shelf BC | 96,373 | 0.21 | 2.17 | 0.03 | 0.29 |
| 3. Southern Shelf BC | 28,313 | 0.09 | 3.11 | 0.01 | 0.34 |
| 4. Strait of Georgia | 8,664 | 0.04 | 4.56 | 0.05 | 5.31 |
| 5. Western Arctic | 526,309 | 1.11 | 2.11 | 0.44 | 0.84 |
| 6. Arctic Basin | 250,178 | 0.45 | 1.78 | 0.02 | 0.08 |
| 7. Arctic Archipelago | 243,425 | 0.48 | 1.97 | 0.02 | 0.06 |
| 8. Eastern Arctic | 757,226 | 1.80 | 2.38 | 0.14 | 0.19 |
| 9. Hudson Bay | 1,234,257 | 3.03 | 2.46 | 1.29 | 1.04 |
| 10. NL Shelves | 820,462 | 1.95 | 2.38 | 0.34 | 0.41 |
| 11. Gulf of St Lawrence | 235,541 | 0.75 | 3.18 | 2.31 | 9.79 |
| 12. Scotian Shelf | 234,888 | 0.61 | 2.59 | 0.21 | 0.90 |

Notes: OC = Organic carbon; NL = Newfoundland-Labrador.

## 3.6 Estimated organic carbon accumulation rates

Organic carbon accumulation rates were estimated from combining mapped products of sediment
mass accumulation and organic carbon content (Fig. 2). Estimated values ranged from $3.5 \times 10^{-6}$
to 76.9 g m⁻² y⁻¹ with a mean of 1.1 ± 2.8 g m⁻² y⁻¹ (± SD). The majority of the model domain was
estimated to have low accumulation rates with values < 0.5 g m⁻² y⁻¹ (Fig. 14). Highest
accumulation rates were restricted to the East coast of Canada across the Gulf of St Lawrence
and in nearshore areas of the Bay of Fundy (Fig. 14). Other areas with relatively high
accumulation rates were confined to near coast areas including the Salish Sea and some fjords
and inlets in the Pacific west coast, as well as near coast areas in Hudson Bay, Foxe Basin and
the Beaufort Sea in the Arctic (Fig. 14). Cell specific upper and lower 95% CI bounds are shown
in Figure D6. On average the upper CI bounds were 88% higher than the mean prediction, and
the lower CI bounds 47% less than their means. Overall, the total accumulation of organic carbon
across the model domain is estimated with a mean of 4.9 Mt y⁻¹ with a 95% confidence interval of
2.6 – 9.3 Mt y⁻¹. In contrast to the organic carbon standing stock, total accumulation between
bioregions was not strongly related to the total areal extent. The Gulf of St Lawrence was



estimated to contain both the largest total annual organic carbon accumulation and the highest
accumulation per unit area (Table 2). The Strait of Georgia was estimated to have the second
highest accumulation rates per unit area, but low total carbon accumulation due to its small area
(Table 2). The Hudson Bay bioregion also included a large proportion of the organic carbon
accumulation across the model domain with the second highest total accumulation value and the
third highest mean per unit area (Table 2).

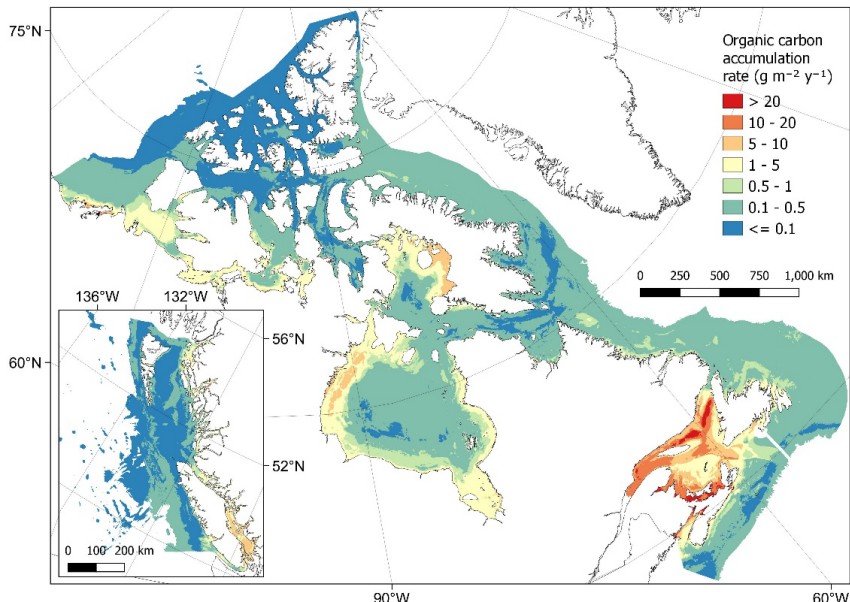

**Figure 14. Estimates of organic carbon accumulation rate (g m⁻² y⁻¹) across the Canadian continental margin.**
The main plot shows the Arctic and Atlantic regions with the Pacific region inset. The continuous variable is shown
displayed in discrete colour bands to improve visualisation of highly right skewed data. The 95% confidence interval
bounds around the predicted means are shown in Figure D6. Labels indicating the locations of different areas
mentioned within the text are shown in Figure A4. Country outlines from World Bank Official Boundaries, available at
https://datacatalog.worldbank.org/search/dataset/0038272.

### 3.7 Rock substrate distribution case studies

As the predictive maps produced in this study rely on physical sediment samples alone, they are
unlikely to produce valid estimates for areas of bedrock - i.e. estimates of zero sediment carbon
density and accumulation where bedrock is located. On the Scotian shelf (bioregion 12),
correcting our predictive maps with a predicted bedrock distribution map (Fig. E1) reduces total

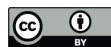



organic carbon stock estimates in this region by between 7.5 – 7.6% leading to a value of 0.56 Gt
(95% CI 0.33 – 0.87 Gt), and reducing total accumulation by 12.7 - 15.9% to a total of 0.18 Mt y$^{-1}$
(95% CI 0.08 – 0.44 Mt y$^{-1}$). For the Pacific British Columbian marine region (bioregions 1-4),
assigning zero values to areas covered by a predicted bedrock distribution map (Fig. E2) would
reduce our estimates by 8.5 - 9.0% to a total of 0.44 Gt (95% CI 0.26 – 0.69 Gt) for organic carbon
stock and reducing by 13.8 – 15.3% to a total of 0.08 Mt y$^{-1}$ (95% CI 0.03 – 0.23 Mt y$^{-1}$) for organic
carbon accumulation.

**4. Code and data availability**
All mapped products as shown in Figures 5, 8, 11, 12, 13 and 14 have been made available as
georeferenced     TIFF     files     in     the     Borealis     data     repository     at
https://borealisdata.ca/privateurl.xhtml?token=7bb00f1e-2ce3-400c-955d-e8e0d4fe3080
(Epstein et al., 2023). This includes the mean predictions as well as the cell-specific 95%
confidence interval bounds as shown in Appendix D. The repository also contains all data collated
within the systematic data review of organic carbon content and the georeferenced TIFF files from
the rock distribution case studies (Appendix E). Additionally, all the associated code used for data
manipulations, model building and predictive mapping can also be found within the above
repository.

**5. Discussion**
Using best available data, we have produced the first national assessment of organic carbon in
surficial seabed sediments across the Canadian continental margin, estimating the standing stock
in the top 30 cm to be 10.7 Gt (95% CI 6.6 – 16.0 Gt). Although comparisons to previous global
studies is challenging due to differences in sediment reference depths, mapping resolutions and
total spatial coverage, our estimate falls within a similar range to those previously published (e.g.
2.2 Gt in the top 5 cm (Lee et al., 2019) and 48 Gt in the top meter (Atwood et al., 2020) of the
Canadian EEZ). In contrast to these global studies, the national approach taken here allows for a
more complete data synthesis, a finer spatial resolution, larger spatial coverage of the Canadian
continental margin and spatially explicit estimates of uncertainty; all of which allow for higher
confidence in the predictive mapping products and overall estimates of standing stock. Similarly
to other national and regional mapping studies (Smeaton et al., 2021; Diesing et al., 2017, 2021),





areas of high organic carbon stocks were predominantly predicted to occur in coastal fjords, inlets,
estuaries, enclosed bays and sheltered basins, as well as in deeper channels and throughs (Fig.
13). To put our estimated organic carbon standing stock into context, 10.7 Gt equates to 51% of
the organic carbon estimated to be stored in all Canadian terrestrial plant live biomass and detritus
(both above and below ground), and 34% of soil organic carbon to 30 cm across Canada
(assuming equal distribution of soil carbon in the top 1 m) (Sothe et al., 2022).
Due to a lack of available data, we were unable to undertake a fully independent predictive
mapping exercise for organic carbon accumulation rates on Canadian seabed sediments.
However, our downscaling exercise of a recently published global product on mass accumulation
rates, coupled with the national predictive mapping of sediment organic carbon content, led to an
estimated annual accumulation at the seafloor of 4.9 Mt of organic carbon per year (95% CI 2.6
– 9.3 Mt y$^{-1}$). Given the extent of the model domain (~1.25% of the global ocean), this estimate
again falls close to the range of previous global predictions – i.e. 1.25% of global accumulation at
126–350 Mt y$^{-1}$ is 1.6-4.4 Mt y$^{-1}$ (Keil, 2017; Berner, 1982). Areas of high accumulation were
predominantly restricted to the Gulf of St Lawrence and Bay of Fundy, as well as other near-coast
areas where large river outlets co-occurred with predicted areas of high carbon density (Fig. 10,
13, Supplement).

*Model interpretation and uncertainties*

The two key components of the carbon stock estimates in this study are the predictive maps for
mud content and organic carbon content, which were estimated to have a map accuracy of 60%
and 58% respectively (R$^2$ 0.60 and 0.58). While these values may seem relatively low when
compared to some other related studies (Diesing et al., 2017, 2021; Atwood et al., 2020; Mitchell
et al., 2019), the use of robust, spatially explicit cross-validation to calculate model evaluation
metrics (as we did herein) has been shown to produce significantly more conservative estimates
of map accuracy when compared to frequently used random cross-validation approaches (Ludwig
et al., 2023; Meyer et al., 2019) such as those used in both the global seabed carbon stock studies
discussed above (Atwood et al., 2020; Lee et al., 2019). Within this study, we also calculated cell
specific confidence interval bounds to give spatially explicit estimates of uncertainty. While there
are many ways to calculate model uncertainty, therefore making comparisons between studies
challenging, the uncertainty in carbon density calculated here (CI 39-54% either side of the mean)
is close to those found within similar regional (Diesing et al., 2021; 58%) and global studies (Lee



et al., 2019; 49%), both of which predict carbon stocks at significantly coarser resolutions. Our
95% confidence interval bounds for total standing stock (38% lower and 50% higher than the
mean) are also similar to the estimated bounds from the recently published predictive models of
Canadian terrestrial vegetation and soil carbon (a 90% confidence interval 48% either side of the
mean) (Sothe et al., 2022).
Higher map accuracy was estimated for mass accumulation rate ($R^2$ 0.89); however, it is important
to recognise that this only describes the accuracy of our downscaled product to represent the
global spatial product from which data were sampled. This global model was itself estimated to
have an $R^2$ of 0.88 for empirical point data, however this was calculated with traditional random
cross-validation techniques (Restreppo et al., 2021). The estimated values of organic carbon
accumulation rate predicted here should be used with some caution as there is likely significant
uncertainty that is not truly quantified due to the small amount of *in-situ* empirical data from the
Canadian continental margin (Restreppo et al., 2021). The mean confidence interval for organic
carbon accumulation estimated in this study was also very wide at its upper bound (88% above
mean). This is largely due to the highly right skewed distribution of predictions, with a
preponderance of small accumulation rate values, meaning a small absolute increase in
estimated accumulation can have very large proportional effects when compared to the mean.
Even so, the estimates of organic carbon accumulation made here give our current best estimate
for the Canadian continental margin, and while the absolute values may contain high uncertainty,
the spatial patterns between areas across the model domain are expected to have higher
confidence.
Using two case studies from British Columbia and the Scotian Shelf, we estimated that the
distribution of rock substrates could reduce our estimates of carbon stock by approximately 7.5 -
9.0% and carbon accumulation by 12.7 – 15.3% (Fig. E1, E2). As much of the Canadian coastline
is distant from significant infrastructure, extensive surveys of the seafloor are generally lacking,
especially when compared to similar regional carbon mapping studies in northwest Europe (e.g.
Smeaton et al., 2021). It is therefore unclear how representative these case studies are of the
entire Canadian EEZ. Improved data on the presence of bedrock across lesser studied regions
of the Canadian Arctic, Hudson Bay, Gulf of St Lawrence, Newfoundland and Labrador may allow
for the production of a predictive map of bedrock across the Canadian EEZ which would
significantly improve the carbon estimates and spatial predictive maps produced in this study.
Areas of uncertainty which could not be fully quantified include the accuracy and precision of
response data and predictor layers. The response data drive the model construction, and



therefore sampling, processing, or recording errors can propagate into predictions. This is
particularly relevant given the large temporal extent of response data which was required to gain
sufficient coverage for this work (1959-2019). This large duration may also add additional variation
from temporal differences between data, for example from differing anthropogenic drivers on
carbon storage and/or accumulation (Keil, 2017); however, similar temporal extents have been
used in related studies (Atwood et al., 2020; Lee et al., 2019; Seiter et al., 2004) and 72% of the
organic carbon data within this study were sampled after 1980 and 55% after 2000. Within the
response data, assumptions and/or predictions were also required regarding the distribution of
mud and carbon across sediment depths. While standardising for this factor is clearly necessary,
especially when using a wide variety of legacy data, it does add additional uncertainty which would
not be present if widescale standardised sampling methods were employed. The results from this
study do however highlight, that within the top 30 cm of sediment, the spatial location of the
sample is a far stronger driver of organic carbon content than the sediment sampling depth (Table
B1). Most of the predictor variables used in this study are also themselves modelled products,
which contain their own inherent uncertainties and/or interpolations which cannot be fully
quantified here. Additionally, many of the predictor variables have temporal components, and
while the climatological mean of a 12 - 14 year timespan used in this study is expected to produce
variables representative for the study region, they do not completely align with the temporal extent
of the response data which could add further prediction uncertainty.

*Future directions and applications*
Improvements could be made in future iterations of these sediment carbon maps when additional
response data become available. The size of the organic carbon content dataset was relatively
small (2,518 point-samples) given the size of the model domain, so new data could greatly
improve accuracy and reduce uncertainty in predictions. Additionally, wide-spread *in-situ* data on
sediment dry bulk density and sediment mass accumulation rates would reduce the assumptions
needed in using transfer functions and downscaling models; however, large datasets would be
needed to conduct robust independent modelling exercises. There are also improvements to be
made with the development of higher resolution or more accurate predictor layers. This would be
particularly relevant for those variables with coarse resolutions and those which were seen to
have highest importance within our models or from related seabed sediment mapping studies
(e.g. Gregr et al., 2021; Diesing et al., 2017, 2021; Mitchell et al., 2019) - i.e. wave velocities,
suspended particulate matter, exposure, current velocities and oxygen concentrations. Further



validation and refinements could also be supported by numerical biogeochemical modelling products where the organic carbon densities and/or accumulations are mathematically estimated based on oceanographic, climatological and benthic conditions, including the potential to incorporate predictions under different future climate scenarios (Ani and Robson, 2021).

The organic carbon predictive mapping products generated here could have many future applications. Regionalisation and prioritisation processes could identify key areas of carbon storage for further research and possible protections (Epstein and Roberts, 2022, 2023; Diesing et al., 2021). There is also potential to combine these mapped products with spatial data on human activities occurring on the seafloor to consider potential management implications, such as controlling the levels of impactful industries (e.g. mobile bottom fishing, mineral extraction, energy generation) in high organic carbon storage/accumulation areas (Clare et al., 2023; Epstein and Roberts, 2022). The mud content predictive maps may also have applications for marine planning more widely, being a strong driver of the biological habitat type and sensitivity. Overall, these data have wide-scale relevance across marine ecology, geology and environmental management disciplines, however, the use of these products should always consider the discussed uncertainties and quantified confidence interval bounds of predictions. As with all large-scale mapping exercises, continued *in-situ* empirical data collection is needed for improved accuracy of mapping seabed carbon stocks and accumulation rates across Canada.





**6. Appendices**
**Appendix A. Distribution of response data**

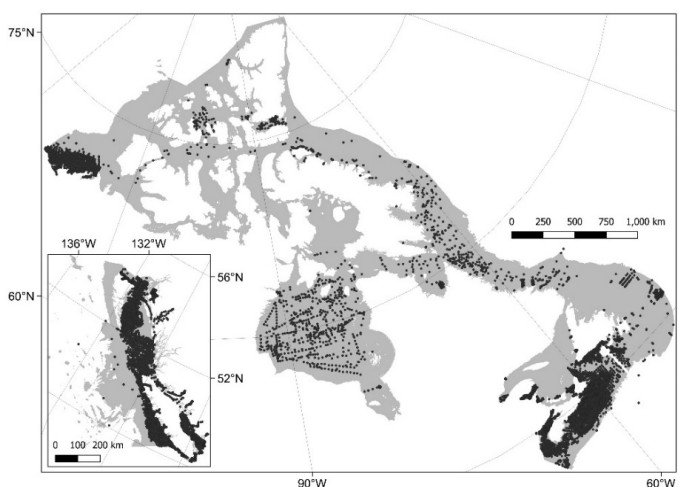


**Figure A1. Map showing the distribution of mud content samples across the model domain.**

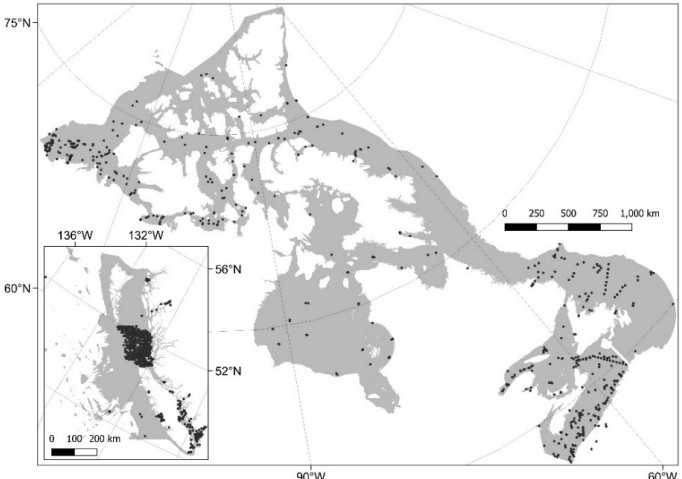


**Figure A2. Map showing the distribution of carbon content samples across the model domain.**

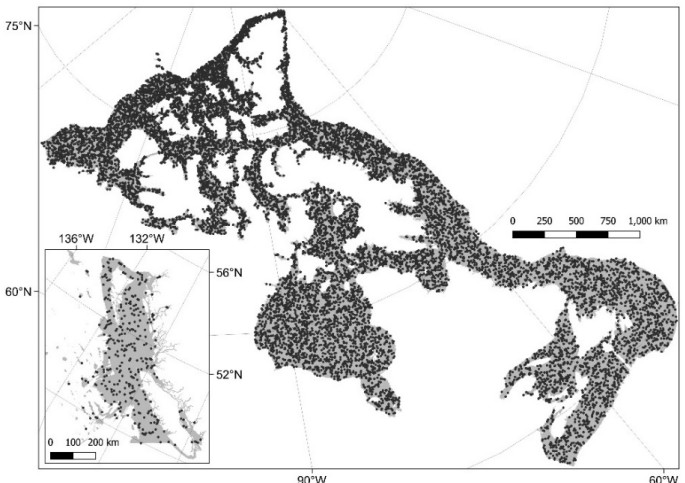


**Figure A3. Map showing the distribution of random-stratified sampled sediment mass accumulation rates across the model domain.**

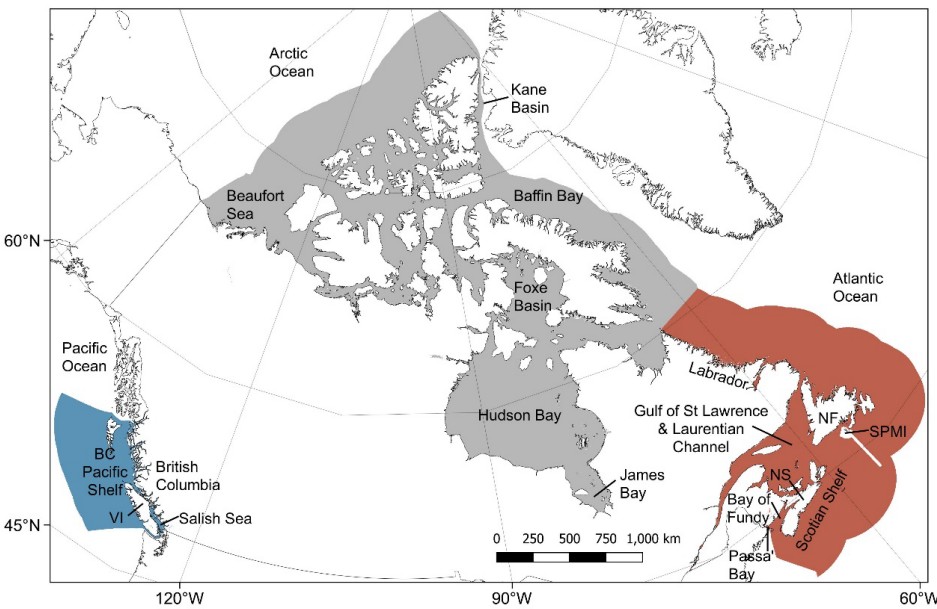


**Figure A4. Map indicating the locations of different areas which are mentioned within the text.** The Canadian Pacific (blue), Arctic (grey) and Atlantic (red) regions are shown with labelled locations overlayed. BC = British Columbia; Passa' Bay = Passamaquoddy Bay; NS = Nova Scotia; NF = Newfoundland; SPMI = St Pierre and Miquelon. The locations are for guidance only and do not represent the entire extent or exact location of a given area. Country outlines are derived from World Bank Official Boundaries, available at https://datacatalog.worldbank.org/search/dataset/0038272.





**Appendix B. Organic carbon sediment depth modelling results**
There was a significant effect of sampling depth on the organic carbon content in seabed
sediments ($\chi^2$ = 1400.9, p < 0.001). While sample ID explained most of the variation between
sub-sample carbon contents, the sampling depth was also a significant factor (Table B1). Carbon
content decreased with increasing sampling depth (Fig. B1). The rate of carbon content decline
generally decreased with increasing depth into the sediment, however uncertainty in this trend
increased within deeper sediment layers (Fig. B1).

**Table B1. Results from the generalised additive mixed model between the carbon content of marine sediments**
**and sampling depth**. A basic generalised additive mixed model with a scaled-t distribution was constructed for carbon
content in sediment sub-samples with sample ID as the random factor and sampling depth as the fixed factor.

| Spline | Type | edf | Res. df | $\chi^2$ | Deviance explained | p |
|---|---|---|---|---|---|---|
| Sampling depth (cm) | Cubic | 4.28 | 5.36 | 2299 | 1.1% | < 0.001 |
| ID | Random | 181.94 | 182.00 | 715046 | 86.9% | < 0.001 |

Notes: edf = Effective degrees of freedom. Res. df = Residual degrees of freedom

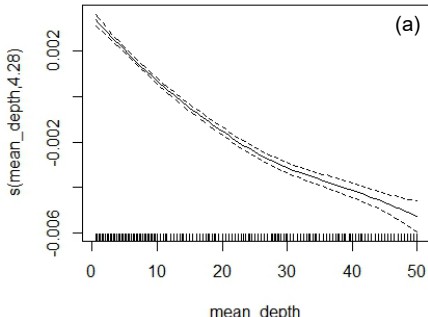
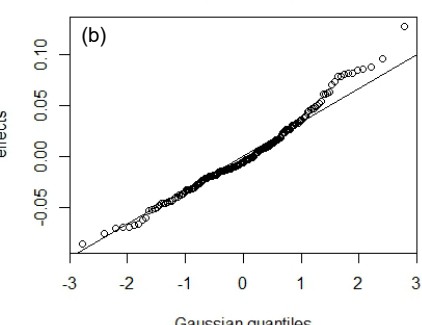

**Figure B1. Regression splines indicating the effect of sediment sampling depth (a) and sample ID (b) on the**
**organic carbon content in seabed sediment sub-samples.**

The predicted mean effect of sediment depth on carbon content was extracted from the model
and converted into a transfer function which states the expected ratio between the cumulative
carbon content at 30 cm compared to any given sampling depth (Figure B2). The ratio ranged
from 89.3% when only measuring the sediment surface, to 93.7% if measuring the carbon content
across the top 10 cm, and by 25 cm was approaching equilibrium at 98.8%.






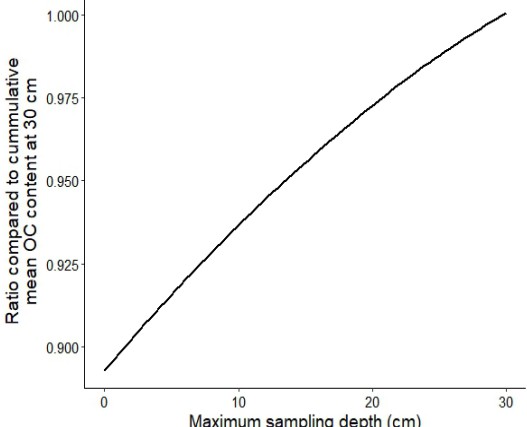


**Figure B2. Transfer function for cumulative mean organic carbon (OC) content at 30 cm sediment depth.** Using
a generalised additive mixed model an estimated transfer function was constructed to standardise the cumulative mean
carbon content at any given depth to an expected value at 30 cm.

**Appendix C. Results from random forest cross-validation structure selection**

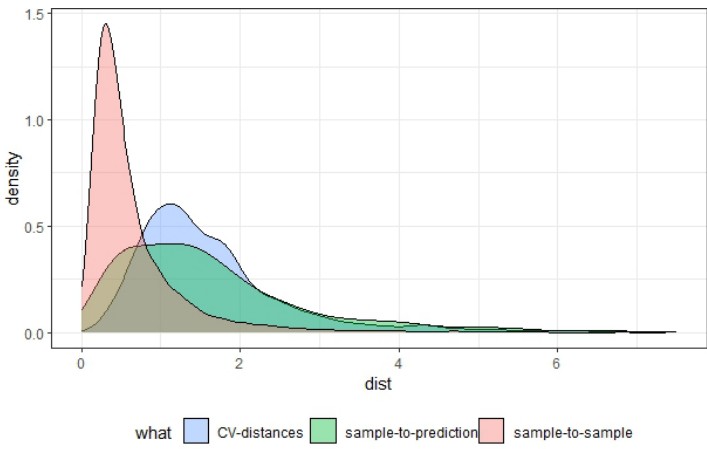


**Figure C1. Multivariate nearest-neighbour distance density plot for mud content data with the optimal number**
**of spatial k-means clusters across cross validation (CV) folds.** Frequency of nearest neighbour distances (x-axis)
is shown for sample-to-sample distance (red), sample-to-prediction distance (green) and CV fold analysis-to-
assessment distance (blue). An optimal number of 35 clusters was selected to due close overlap between the CV-
distance and sample-to-prediction curve.



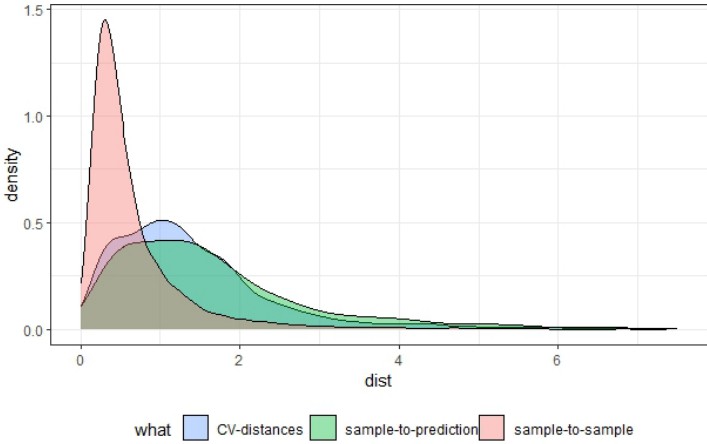


**Figure C2. Multivariate nearest-neighbour distance density plot for mud content data with a partially repeated spatial-random mixture method for cross validation (CV) folds.** Frequency of nearest neighbour distances (x-axis) is shown for sample-to-sample distance (red), sample-to-prediction distance (green) and CV fold analysis-to-assessment distance (blue). Due to the optimal spatial k-means clustering showing poor overlap at lower multivariate distances (Fig. C1), a 1% random sample without replacement was added to each fold.

1121

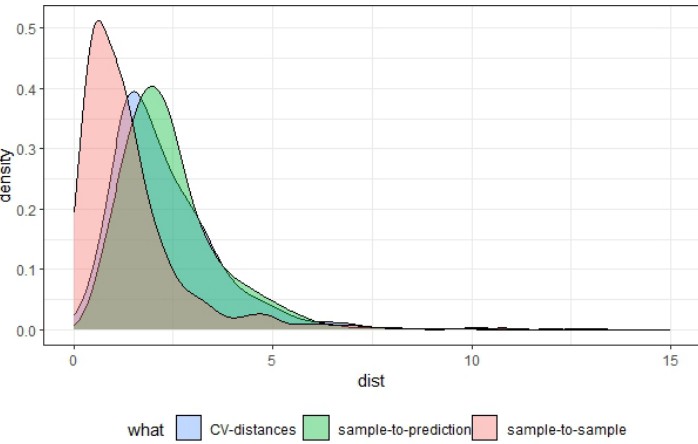

1122

**Figure C3. Multivariate nearest neighbour distance density plot for organic carbon content data with the optimal block size across cross validation (CV) folds.** Frequency of nearest neighbour distances (x-axis) is shown for sample-to-sample distance (red), sample-to-prediction distance (green) and CV fold analysis-to-assessment distance (blue). An optimal block size of 100 km was selected to due close overlap between the CV-distance and sample-to-prediction curve.



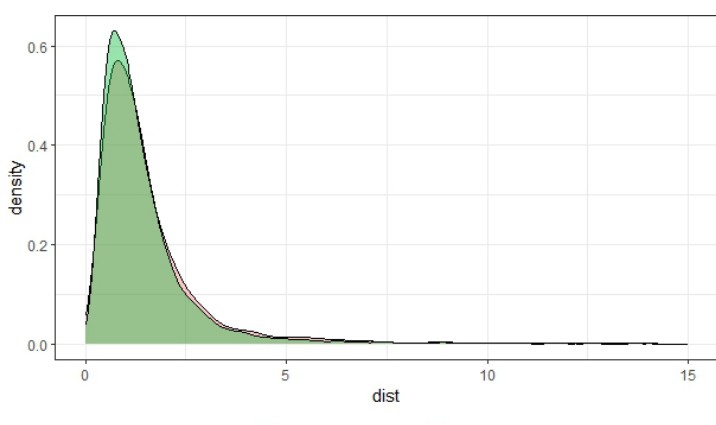


**Figure C4. Multivariate nearest neighbour density plot for sediment mass accumulation rate data.** Frequency of
nearest neighbour distances (x-axis) is shown for sample-to-sample distance (red) and sample-to-prediction distance
(green). The close overlap indicates that random cross-validation will produce valid results.

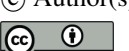
**Appendix D. Cell-specific confidence interval bounds for predictive sediment maps**

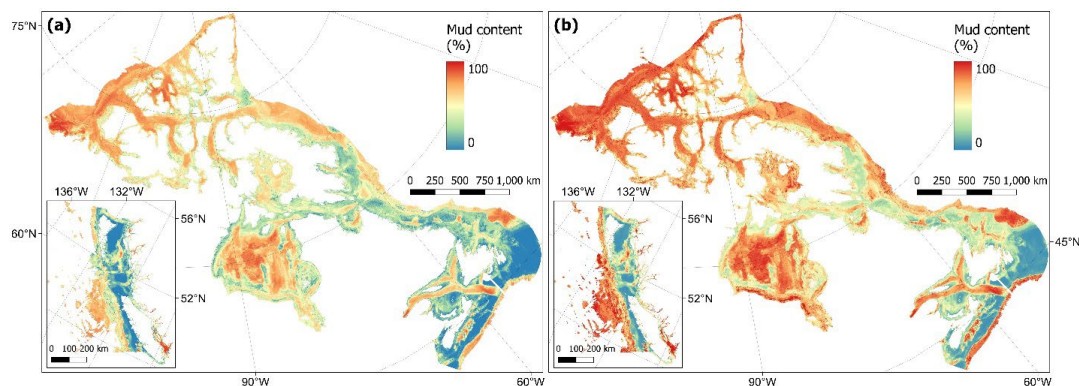


**Figure D1. Estimated lower (a) and upper (b) bounds of the 95% confidence interval for predictions of mud**
**content (%) in subtidal marine sediments across the Canadian continental margin.** Within each panel the main
plot shows the Arctic and Atlantic regions with the Pacific region inset.

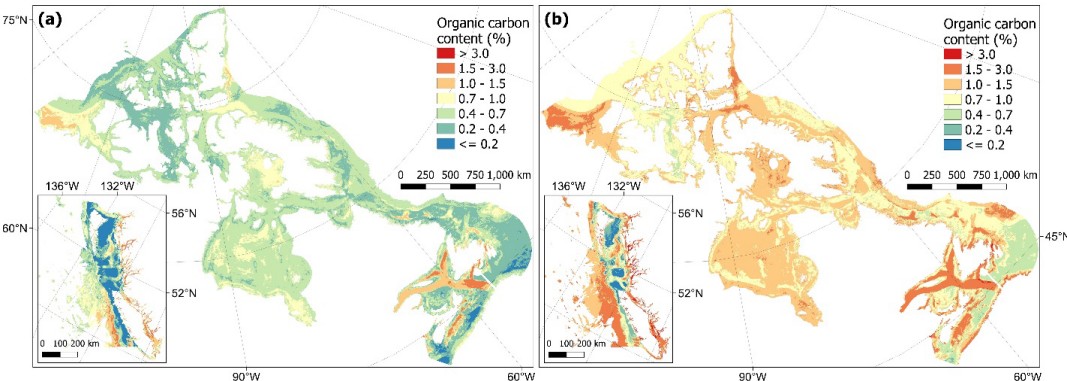


**Figure D2. Estimated lower (a) and upper (b) bounds of the 95% confidence interval for predictions of carbon**
**content (%) in subtidal marine sediments across the Canadian continental margin.** The continuous variable is
shown in discrete colour bands to improve visualisation of highly right skewed data. Within each panel the main plot
shows the Arctic and Atlantic regions with the Pacific region inset.



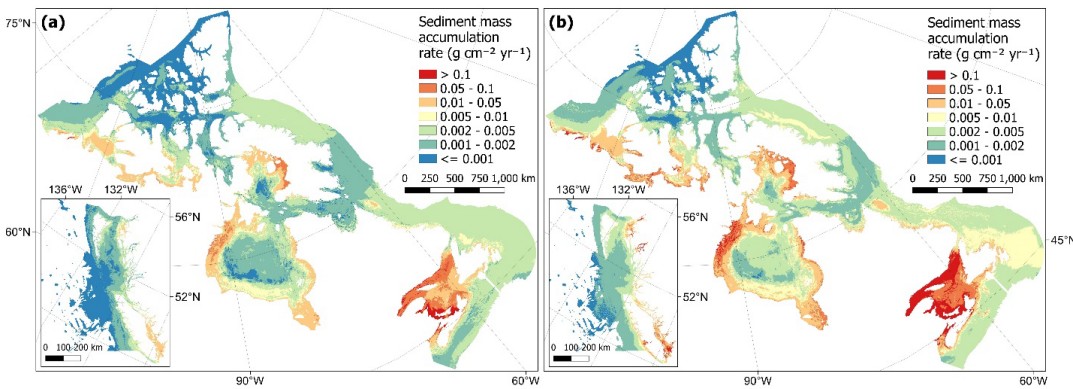

**Figure D3. Estimated lower (a) and upper (b) bounds of the 95% confidence interval for predictions of mass accumulation rate (g cm$^{-2}$ yr$^{-1}$) on subtidal marine sediments across the Canadian continental margin.** The continuous variable is shown in discrete colour bands to improve visualisation of highly right skewed data. Within each panel the main plot shows the Arctic and Atlantic regions with the Pacific region inset.

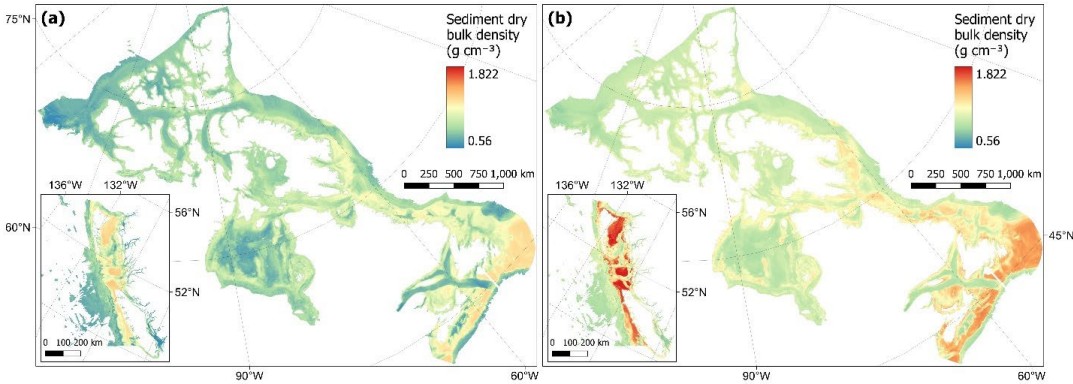

**Figure D4. Estimated lower (a) and upper (b) bounds of the 95% confidence interval for predictions of dry bulk density (g cm$^{-3}$) of subtidal marine sediments across the Canadian continental margin.** Within each panel the main plot shows the Arctic and Atlantic regions with the Pacific region inset.



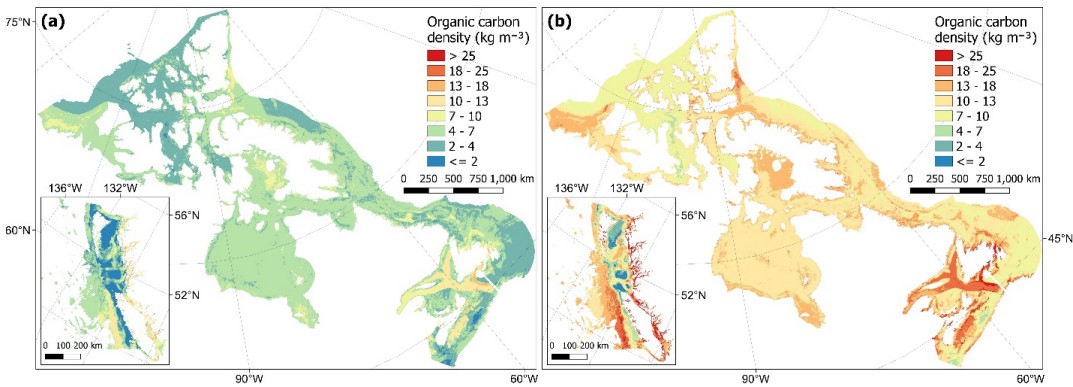

1156

**Figure D5. Estimated lower (a) and upper (b) bounds of the 95% confidence interval for predictions of organic carbon density (kg m⁻³) in subtidal marine sediments across the Canadian continental margin.** The continuous variable is shown in discrete colour bands to improve visualisation of highly right skewed data. Within each panel the main plot shows the Arctic and Atlantic regions with the Pacific region inset.

1161

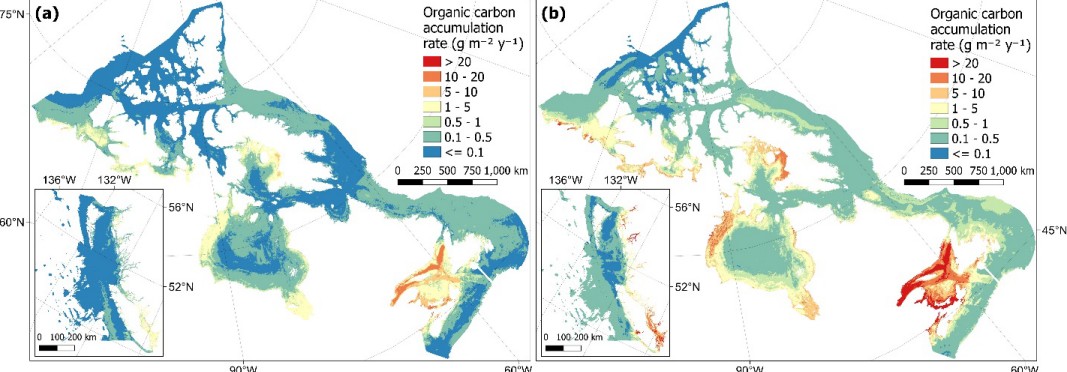

1162

**Figure D6. Estimated lower (a) and upper (b) bounds of the 95% confidence interval for predictions of organic carbon accumulation rates (g m⁻² y⁻¹) on subtidal marine sediments across the Canadian continental margin.** The continuous variable is shown in discrete colour bands to improve visualisation of highly right skewed data. Within each panel the main plot shows the Arctic and Atlantic regions with the Pacific region inset.





**Appendix E. Bedrock distribution case studies**

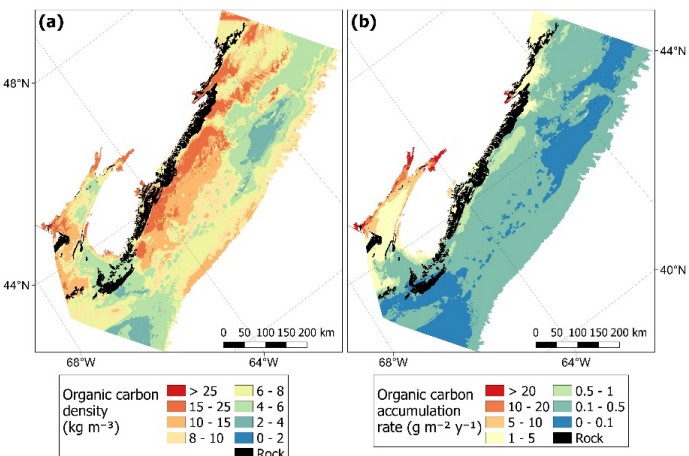


**Figure E1. Predicted mean values of organic carbon density and accumulation rates within the Scotian Shelf**
**overlayed by the estimated distribution of rock substrates.** Data on the estimated distribution of rock on the
seafloor across the Scotian Shelf Bioregion is taken from Philibert et al. (2022).


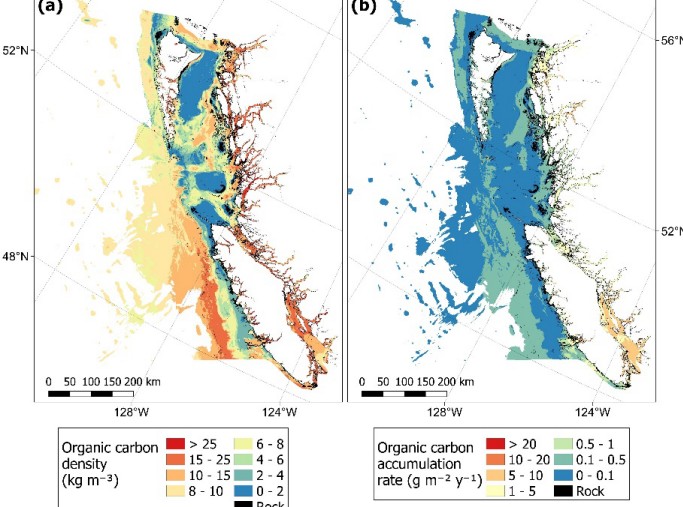


**Figure E2. Predicted mean values of organic carbon density and accumulation rates within the British**
**Columbia EEZ overlayed by the estimated distribution of rock substrates.** Data on the estimated distribution of
rock on the seafloor across the British Columbian continental margin is taken from Gregr et al. (2021).



## 7. Author contributions

JKB and SDF secured funding and led the management of this project. GE, SDF and JKB conceptualised this study. GE, DH, AP, CP & PGM collated the data. GE developed the model code and performed the investigations with input from SDF and JKB throughout. GE prepared the manuscript with contributions from all co-authors.

## 8. Competing interests

The authors declare that they have no conflict of interest.

## 9. Acknowledgements

We would like to thank Randy Enkin, Sarah Paradis and Genevieve Philibert for providing data towards this work. We also greatly appreciate advice given across various stages of the processes from Cooper Stacey, Markus Diesing, Ashley Park, Nadja Steiner, Diane Lavoie, Amber Holdsworth, Sophia Johannessen, Michael Li, Kate Jarret, Javier Murillo-Perez, Ellen Kenchington, Emily Rubidge and others within the Department for Fisheries and Oceans, and Natural Resources Canada. We would also like to thank Jennifer McHenry, Matt Csordas and Brian Timmer for their ideas in trouble-shooting discussions. This research was enabled in part by support provided by BC & Prairies Digital Research Infrastructure and the Digital Research Alliance of Canada (alliancecan.ca).

## 10. Financial support

This work was funded by an Natural Sciences and Engineering Research Council (NSERC) Alliance partnership grant #ALLRP571068 – 21 to JKB, and is publication #001 of Blue Carbon Canada. GE is also supported by a Mitacs-Accelerate Fellowship, jointly funded by Oceans North. PGM also gratefully acknowledge the financial and logistic support of grants from NSERC including a Discovery Grant (rgpin 227438-09) and Climate Change and Atmospheric Research Grants (VITALS - RGPCC433898 and the Canadian Arctic Geotraces program - RGPCC 433848), as well as support from the Marine Environmental Observation, Prediction and Response Network (MEOPAR) Prediction Core.





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
