# Peer review of "in surficial sediments of the Canadian continental margin"

_Earth System Science Data, 2023_

## Referee Comment (RC1)

Review of "Predictive mapping of organic carbon stocks and accumulation rates in surficial sediments of the Canadian continental margin" by Epstein et al.

General comments (section evaluating overall quality of the preprint)

Overall, this manuscript is in great condition, and I recommend publication. There are a couple of things about this manuscript I do not agree with but in all, I would say minor edits are needed. This publication builds off previous efforts to predict seafloor properties including organic carbon content, mass accumulation rate, and mud content. These results are then incorporated into basic calculations to estimate density and accumulation rates. This manuscript focuses primarily the Canadian EEZ and the final results are relevant to a wide range of interests. The methods this paper uses are sound and robust however there are several issues I have which I outline briefly here.

Increasing spatial variability does not decrease uncertainty. There are several times the author makes this point. As an example, you can assume the median and/or mean of an observational dataset and arrive at a residual value (obs-pred) similar to that of the prediction. Further, statements made regarding the term "in-situ" data should be removed. None of this data was truly *in-situ* data as this data was pulled shipboard and processed ex-situ (i.e., on-deck). Overall, the text is very well written. However, the length and level of detail in the manuscript is at times very overwhelming (e.g., methods). The manuscript could benefit from being shortened to add to clarity for the reader. Certain details presented in the manuscript are more appropriate for the supplemental material (e.g., modules used). A major issue I would like to see addressed in a revision is the MAR prediction. It does not make sense to me the way this observed data was generated (more details in the specific comments below). Further, more specific comments about the manuscript I would like to see addressed below.

Specific comments

Line 29: Consider rearranging this sentence for clarity to say density and accumulation for mud content, sediment dry bulk density, and organic carbon content. Otherwise it reads differently, I at first questioned the different between organic carbon content, organic carbon density, and organic carbon accumulation.

Line 38: See general comments regarding the term of *in-situ*

Figure 1. Why the red and grey parts? After reading the manuscript I did do not have a full understanding of what this is.

Quite a few predictors are up to several orders of magnitude upsampled to be at the resolution of prediction. By upsampling you are not adding new information (variance) therefore your prediction will not be any more accurate and/or uncertain. Consider this in relation to general comments regarding spatial uncertainty and variability.

Line 197: Why do annulus and square windows? And why these different annulus widths? Are these selections related to processes or arbitrary? Do you find that the square windows cause artifacts on the final predictions?

Lines 201-203: You do this to several sets of grids. How different are these grids after this process? This is a lot of manipulation; how do you think this affects the final prediction? Particularly for those predictions that place emphasis on these as predictor grids?

Line 207: It would be beneficial to use same terms (shelteredness, exposition) in the text as in the Table 1.

Line 296: What is the lowest vertical cell? How often was this close enough to the sea bottom? Enough to be accurate to constrain what the value at the seafloor would actually be?

General note about methods/predictor grid processing: The paper would benefit from figure(s) outlining what was done perhaps within the supplemental material. In some cases, there was similar processing for various sets of grids so reusing the same flow diagram and referencing it might provide clarity. The level of detail given throughout the manuscript is great, however it is cumbersome to understand at times.

Line 273: Why three different ocean circulation models? State explicitly. Additionally, when you put all these grids together do you see artifacts?

General comment about grids for Section 2.3.4: After performing all this processing, are these grids any better than the global circulation model estimates? I understand that the grids are better resolution when taken on time slice or over particular areas (you mention explicitly nearshore areas) however, is some of that fidelity for other regions stripped by the processing techniques?

Section 2.4: This is grain size not composition.

Line 434: What about Hayes et al., 2021 (https://doi.org/10.1029/2020GB006769) and the CASCADES Martens et al., 2021 (https://doi.org/10.5194/essd-13-2561-2021) dataset?

Section 2.5.2: I am confused some by this section. Why the upper 30 cm? Why not select the upper $n$ cm that are controlled by the data depth distribution? 30 cm can account for long periods of geologic time in some cases (lower sed rates). Further, by approximating a mean decay function are you starting to incorporate effects of degradation? What uncertainty are you introducing by this entire processing? This seems a lot of unnecessary data processing for observed data.

Section 2.7: I do not agree with using this data as observed data. Why not just upsample the prediction? The observed data is not real and thus should not be used in this manner. No prediction even if the predictors are fantastic will give you the correct result if the observed data is not solid. Further, why sample it spatially randomly and not at the same locations you have observed data for OC? I think you are arbitrarily making your predictions better by randomly sampling as you are covering the feature (predictor) space more uniformly. We find the same phenomenon occurs when performing this on synthetic data. The synthetic data will always outperform as it covers the feature space more uniformly and predictors are inherently tied to spatial phenomenon. I would suggest if you (although I do not agree with it) are to use this MAR data like this then you should sample it spatial and with as many samples as other datasets (e.g., OC) that have been sampled. This will represent the natural (but unfortunate) bias that occurs in marine sediment sampling.

Line 530: This is not completely spatial explicit if you are adding x and y coordinates as predictor variables?

Line 607: Why change the *mtry* and *min_n* number and not the number of trees?

Line 626: How did you merge these datasets? Were there edge effects?

What about using confidence intervals on non- normally distributed data?

Section 2.9: Why are you calculating dry bulk density in several different ways? Why are you calculating porosity with three different equations? This section lacks clarity and I left feeling confused about what was done.

Did you do anything to avoid collinearity between the predictor grids that were selected?

Figure 4 caption: Indicate more clearly which parts of these is ALE (distros drawn by line) and which is a rug plot (bottom)

On what grounds (citations?) were certain predictors selected as *a priori* for input to the various predictions? (e.g., Fig 3)

Line 788: is this a typo? RMSE of 0.09 arcsin{%OC} also what about line 808 "RMSE of 0.206 log10{g cm-2yr-1}." Why say it like this?

---

## Referee Comment (RC2)

I truly enjoyed reading this manuscript and believe it to be in very good shape overall. The study was well-designed, and the manuscript well-written. I recommend publication generally, but would like to raise a few points that the authors and editor should consider before proceeding. I am not an expert on marine carbon, and I am not a statistician, but I have several questions about those components, and can additionally provide perspective on the sediment and geospatial modelling components. I hope these comments are valuable.

**General comments**

The Introduction provides a good amount of background on the motivations for this work. It appears to sufficiently cover the current state of knowledge regarding broad scale marine carbon modelling. I enjoyed reading it, and have nothing major to suggest.

The level of detail describing the predictor variables is quite good. I believe that the reader has enough information here to go ahead and extract similar data if they wanted. That is great.

The effort undertaken to extract useful data from the literature is very impressive. This is one of the best parts of the manuscript to me. It is highly laudable to attempt an exhaustive compilation of existing data from the published literature, rather than ignoring large amounts of previous work in favour of expedient downloads from one or two large repositories. I would like to see more of this for such regional mapping projects.

The Methods section is a bit long, but this may be acceptable given the scope of the project. If the authors are able to improve the conciseness of this section it would be nice for the reader, but maybe that is not possible. Food for thought.

I am not sure if the method employed to calculate confidence intervals for various location specific carbon parameter predictions, and also for the estimates of carbon stocks across the entire Canadian margin, is appropriate. The same logic is applied in many places to derive confidence intervals for a parameter that is based on confidence intervals from a prior model prediction, which I discuss in a bit more detail below. This is my one major concern with the manuscript. Happy to be corrected if I am mistaken here, but please see my comments on the subject below.

**Specific comments**

1) **Page 5.** The colour ramp for the bathymetry is unorthodox. This is not necessarily a problem, bit it looks a bit odd. Is there any reason why dark blue is used for shallow and light yellow for deep?

2) **Page 10-11.** Suggest making it clear here that "SPM" refers to suspended particulate matter. It is not formally defined.

3) **Page 11, L 245.** Consider using the word "disparity" rather than "dissimilarity" here.

4) **Page 19**. It is not currently clear to me why the MAR netCDF had to be sampled and modelled when you already had full-extent spatial predictions of MAR. Why not just resample/interpolate these to the appropriate resolution and projection rather than modelling them? They have already

been modelled once in the first place – it seems like a second round of modelling may just compound the error associated with these predictions.

5) **Page 20, L 543-560.** I found this section a bit difficult to follow. It would help to be very consistent with the terminology here. For example, terms like "analysis data" and "assessment data" are used. Are these the same as "training" and "test" data, which are used later on? I am pretty sure these are the same thing, but consistency would be good, especially because the validation procedure was fairly involved.

Also in this paragraph, you write that "*…This function creates density plots of nearest neighbour distances in multivariate predictor space between all response data…*" By definition, aren't nearest neighbour distances the distance between a data point and its nearest neighbour (in multivariate space), not between all data points?

Relatedly, could you make it clear what these multivariate distances are? Were the predictors normalized before calculating (Euclidean) distances? That would have a very large impact on the distance calculations. This is also not clear from the appendix figures, which are just labelled "dist".

6) **Page 21-22, L 586-605**. Can you clarify how the a priori feature selection was accomplished? You write that variable importance was calculated using a "*basic random forest model with all training data and predictor variables…*" Does this imply that you used the out-of-bag samples to estimate variable importance, or was it estimated on the training data?

7) **Page 22**. I found this section also hard to follow, and again, believe it would be helpful if the terminology was consistent. I'm assuming the "training" and "test" sets here are the "analysis" and "assessment" data from above? This may not be obvious to some readers because "assessment" or "validation" data are often used for model tuning, while a separate "test" set is withheld to calculate the final accuracy.

I'm having trouble understanding the hyper-parameter optimization and validation design described on this page. Can you confirm that separate models were built and tuned for each fold, and were also validated using the same design, and that the out-of-bag (OOB) samples were not used anywhere in this procedure? If that is the case, why not use the OOB samples to tune hyper-parameters, then use the folds for validation? Tuning hyper-parameters on the validation data can slowly lead to a form of overfitting.

8) **Page 22, L 615**. Sorry, but in the same vein the terminology is confusing to me here. You write that:

*"Overall model performance metrics (RMSE and R2) were then calculated using the predictions across all CV folds with optimal hyperparameters and the last-fit; while predictor variable importance was calculated by fitting an additional model across all training data using optimal tuning parameters and the importance calculated through permutation."*

Does this imply that, while performance was calculated using CV, the variable importance was calculated using the training data (i.e., the "analysis" data)? When you say permutation,

does this mean you permute the predictor variables and measure the impact of the model predictions on the training fit then? Or are the OOB data being used here? How were these "optimal tuning parameters" selected if hyper-parameter tuning was performed per fold?

9) **Page 22**. These paragraphs appear to imply that you predicted each of the 10 models from the CV across the entire domain, correct? Or was it a single model fitted "*across all training data*" from above? In either case, what does it mean that the data were randomly split into 150 samples, predicted, then merged to produce the final raster layer? What data were split and why is it random? Please clarify this.

10) **Page 23, L 658**. "Fourth", not "forth".

11) **Page 24, L 675-678**. Does this mean that the final bulk density estimates were the mean of the 14 individual estimates? How did you get standard errors after aggregating these predictions? Did you just use the standard deviation of the 14 predictions? How then can you get a 95% confidence interval? This needs clear explanation. As far as I can tell, you have a 95% CI for four of the transfer functions... how could you possibly pool these, in addition to the other estimates?

12) **Page 24, L 683-685**. You use the 95% CI bounds of the %OC predictions and the MAR predictions to calculate a 95% CI for OC accumulation rates. Why is that appropriate? For example, the upper bound of the %OC and MAR intervals implies you are 97.5% sure the mean value at a given location should be less than that value. Your calculation implies that, given 97.5% confidence that the mean OC% is < x, and given 97.5% confidence that the mean MAR is < y, you are 97.5% sure that the mean OC accumulation rate is < xy. It may be worthwhile to consult a statistician here, but I don't think that's the proper way to calculate a confidence interval. Would you not need some information about the joint distribution of these variables? I'm not a statistician; if I am wrong, please provide some source for this approach.

This has very important implications for your calculations across the entire model domain. Is it appropriate to take the upper and lower confidence bounds of multiple variables, multiply them together, and aggregate those to arrive at 95% confidence that the total OC content across the entire shelf is within a certain range?

13) **Page 26 and surrounding**. A minor comment, but it would be nice if the fonts of all plot elements (e.g., axis labels) were consistent across figures.

14) **Page 28**. The high importance of the predicted mud content raises an important question for me regarding the potential for "data leakage" in your validation. Can you confirm that none of the %OC measurements came from data points where the mud content was also measured and used for those models? It seems possible that this could have occurred since many of both data points were sourced from NRCan, and that there were many more mud samples than %OC in your dataset.

If the mud data points were used to create map predictions at all locations where you have %OC measurements, yet the %OC measurements were sourced from data points that occur in the mud dataset, it is possible that the test data from your cross-validation folds are "leaking" information

to the training data via the predicted mud layer. This would occur due to expected correlations between the mud and organic carbon content if they were obtained from the same sample, and not assigned the same CV folds. It would potentially inflate both the estimates of importance of the mud layer, and the estimates of model performance, despite your substantial efforts at ensuring an appropriate CV design.

15) **Page 32**. Similar comment to previous. You are assuming that the 95% CI can be acquired for dry bulk density predictions simply by performing the calculations using upper and lower bounds of the mud predictions. I'm not sure that is the case. I think it is true to say that you are calculating the *likely dry bulk density given the 95% CI bounds of the mud predictions*, but I don't think that is the same as calculating the 95% CI of the dry bulk density.

16) **Page 35**. Same comments as previous regarding the propagation of confidence intervals.

---

## Author Response (AR1)

We thank Dr Lee & Dr Misiuk for their constructive and through review of the manuscript and their generally positive perspective on its quality and appropriateness for publication. We have carefully considered all of their comments and respond to each of them individually below. Please note line numbers in our responses (where relevant) refer to the attached "tracked changes" version of the manuscript.

**RESPONSE TO REFEREE 1 – Dr Lee**

*General comments*

*"Increasing spatial variability does not decrease uncertainty. There are several times the author makes this point. As an example, you can assume the median and/or mean of an observational dataset and arrive at a residual value (obs-pred) similar to that of the prediction."*
**RESPONSE: We accept the referee's criticism of this narrative. Corrections/retractions have been made in the Abstract, Introduction and Discussion sections to remove this potential inaccuracy. See lines 22-25, 105-109, 124-126, 1033-1035.**

*"Further, statements made regarding the term "in-situ" data should be removed. None of this data was truly in-situ data as this data was pulled shipboard and processed ex-situ (i.e., on-deck)."*
**RESPONSE: The use of the term "*in-situ*" has been removed throughout.**

*"Overall, the text is very well written. However, the length and level of detail in the manuscript is at times very overwhelming (e.g., methods). The manuscript could benefit from being shortened to add to clarity for the reader. Certain details presented in the manuscript are more appropriate for the supplemental material (e.g., modules used)."*
**RESPONSE: We appreciate that the original methods section was quite lengthy. We have therefore moved some of the general (or less integral) parts of the methods to an Appendix where possible – now Appendix A. However, the methods section on the "Organic carbon content data" (now Section 2.3) was not altered as it was considered of particular value by referee #2.**

*"A major issue I would like to see addressed in a revision is the MAR prediction. It does not make sense to me the way this observed data was generated (more details in the specific comments below)."*
**RESPONSE: Following these comments (and those below), as well as those from Referee 2, and further discussions with our collaborators who have significant experience of direct empirical measurement and investigation of MAR and CAR in Canadian seabed sediments, we no longer have sufficient confidence regarding the accuracy of the global modelled MAR layer across many parts of Canadian continental margin. We have therefore made the decision to remove the MAR and CAR predictive modelling from the manuscript entirely.**

***Specific comments:***

*"Line 29: Consider rearranging this sentence for clarity to say density and accumulation for mud content, sediment dry bulk density, and organic carbon content. Otherwise it reads differently, I at first questioned the different between organic carbon content, organic carbon density, and organic carbon accumulation."*

**RESPONSE: This sentence has been altered to state 'Predictive quantitative maps of mud content, sediment dry bulk density, organic carbon content and organic carbon density were produced..'. See lines 29-31. The terms "organic carbon content" (meaning % of sediment dry weight) and "organic carbon density" (meaning mass per unit volume of sediment) are standard.**

*"Line 38: See general comments regarding the term of in-situ"*

**RESPONSE: This term has been removed.**

*"Figure 1. Why the red and grey parts? After reading the manuscript I did do not have a full understanding of what this is."*

**RESPONSE: The figure legend of Figure 1 has been updated so this can be more clearly understood. Now stating "The study area spatial maxima (red) was defined using best available bathymetry data and covers the entire sub-tidal portion of the Canadian EEZ….. This is overlayed by the maximum potential modelling extent (grey) which indicates only those areas where data were present for all predictor variables. Due to the distribution of available response data, the final modelling domain was limited to a depth of 2,500 and is indicated with the colour relative to the estimated depth, from 0 (light blue) to -2,500 (black)."**

*"Quite a few predictors are up to several orders of magnitude upsampled to be at the resolution of prediction. By upsampling you are not adding new information (variance) therefore your prediction will not be any more accurate and/or uncertain. Consider this in relation to general comments regarding spatial uncertainty and variability."*

**RESPONSE: We recognize and fully agree with this statement (although a level of additional information/variance is added from chosen interpolation methods). There is some discussion of this assumption/uncertainty already present in the "Future directions and applications" section of the manuscript, however, we agree that further discussion and detail should be added. See additional content in the "Model interpretation and uncertainties" section of the discussion – see lines 1119 - 1124**

*"Line 197: Why do annulus and square windows? And why these different annulus widths? Are these selections related to processes or arbitrary? Do you find that the square windows cause artifacts on the final predictions?"*

**RESPONSE: Vector Ruggedness Measures and Benthic Position Indices are standard measures of benthic terrain. We consider it best practice to use published procedures and packages to construct these metrics which do indeed employ annulus windows for BPI and square windows for VRM. We do not think it is of interest in the reader to discuss these methods in more detail here. The use of multiple window widths to capture both small local features and larger spatial variation in terrain (as mentioned in the manuscript) is**

**also standard practice (e.g. see (Gregr et al., 2021; Maxwell and Shobe, 2022; Mitchell et al., 2019)).**

*"Lines 201-203: You do this to several sets of grids. How different are these grids after this process? This is a lot of manipulation; how do you think this affects the final prediction? Particularly for those predictions that place emphasis on these as predictor grids?"*
**RESPONSE: The aggregation, processing and disaggregation of layers is only conducted on a very small minority of predictors (3 of the benthic terrain features and only the two broadest components of the 6-part exposure proxy calculations – i.e. ~3.4 out of 28 predictors). In all these cases, calculations are being made on the input layer over a very large window surrounding a focal cell to capture broad scale patterns in terrain/distance to shore. For this reason, small changes in native resolution (in this case aggregation from just 200m to 400m) prior to processing should have little difference on the outputs. Further, the random forest models do not show high importance to the terrain metrics, and for the exposure proxy calculations the majority (4 out of 6) of the processing layers are based on a full resolution grid.**

*"Line 207: It would be beneficial to use same terms (shelteredness, exposition) in the text as in the Table 1."*
**RESPONSE: Sentence altered to "a proxy measure for exposure describing the degree of exposition vs. shelteredness" See lines 223-224.**

*"Line 296: What is the lowest vertical cell? How often was this close enough to the sea bottom? Enough to be accurate to constrain what the value at the seafloor would actually be?"*
**RESPONSE: We recognize that this may not have been clear, but the lowest vertical cell in each model is indeed the one which contacts the seafloor. The sentence has been altered to state "For each horizontal cell, data were only retained from the lowest vertical cell within a given position (i.e. the cell which contacts the seafloor)". See lines 311-312.**

*"General note about methods/predictor grid processing: The paper would benefit from figure(s) outlining what was done perhaps within the supplemental material. In some cases, there was similar processing for various sets of grids so reusing the same flow diagram and referencing it might provide clarity. The level of detail given throughout the manuscript is great, however it is cumbersome to understand at times."*
**RESPONSE: The methods section should now be less cumbersome to understand – appropriate parts have been moved to Appendix A and the MAR and CAR sections are no longer present. We do not think that the details of the predictor layer construction could be easily adapted into an intelligible descriptive figure/figures. Further, as the manuscript is already very figure heavy, and already has a large Appendix and Supplement we are hesitant to add more figures.**

*"Line 273: Why three different ocean circulation models? State explicitly. Additionally, when you put all these grids together do you see artifacts?"*
**RESPONSE: Sentence added to start of this section to state "To incorporate best available**

regional evidence, data on the mean surface ice cover, seafloor salinity, temperature and current velocity was collated from three different ocean circulation model products covering different regions of Canada" – see line 288-290. Additionally, in reference to this point we also state "As the BCCM model has higher uncertainty in nearshore and enclosed environments due to its relatively coarse resolution, data were also extracted for the enclosed Salish Sea from the Salish Sea Cast ERDDAP data server"– now found in Appendix A3 (see lines 1212-1213) Further details regarding the consideration of edge-effects have been added see lines 333-337.

*"General comment about grids for Section 2.3.4: After performing all this processing, are these grids any better than the global circulation model estimates? I understand that the grids are better resolution when taken on time slice or over particular areas (you mention explicitly nearshore areas) however, is some of that fidelity for other regions stripped by the processing techniques?"*
RESPONSE: Yes, we believe that the regional/local data we've used provide improvements to the currently available global model outputs. Although we do not think it is of direct relevance (or of interest to the reader) to formally compare these products to the range of available global products, further information regarding the detail and novelty of each of the circulation models can be found in the respective relevant citations. We also do not understand the comment by the referee regarding "fidelity for other regions being stripped by the processing techniques". The methods outlined in this section (now 2.1.4) are specifically designed to maintain fidelity of the original circulation models as much as possible but appropriately applying them to the resolution and CRS of this study.

*"Section 2.4: This is grain size not composition."*
RESPONSE: Title altered to "Sediment mud content data" (now Section 2.2).  However, the term "sediment composition" is still used where relevant in the manuscript (this is a commonly used term when discussing the combined metrics of %mud/sand/gravel… e.g. (Mitchell et al., 2019)).

*"Line 434: What about Hayes et al., 2021 (https://doi.org/10.1029/2020GB006769) and the CASCADES Martens et al., 2021 (https://doi.org/10.5194/essd-13-2561-2021) dataset?"*
RESPONSE: We recognize that our use of large data compilation studies published in the primary literature should be more explicitly stated. We now reference the use of CASCADE Martens et al., 2021 and MOSAIC v2 Paradis et al., 2023 explicitly in this section – see lines 429-430. While Hayes et al., 2021 was not used in this study, we have confirmed that all relevant data contained in Hayes et al., 2021 has been incorporated into this study (likely from other global collations/repositories).

*"Section 2.5.2: I am confused some by this section. Why the upper 30 cm? Why not select the upper n cm that are controlled by the data depth distribution? 30 cm can account for long periods of geologic time in some cases (lower sed rates). Further, by approximating a mean decay function are you starting to incorporate effects of degradation? What uncertainty are you introducing by this entire processing? This seems a lot of unnecessary data processing for*

*observed data."*

**RESPONSE: While understanding your point, due to the large variation in the maximum sediment sampling depth of %OC point samples, some process of standardization across data is necessary. We chose 30 cm as there was good representation of data up to this depth and it is a commonly suggested carbon stock accounting depth for terrestrial soil and marine sediment habitats in both carbon accrediting methodologies and greenhouse gas inventories (IPPC, 2019; VERRA, 2020). Although we may not have fully understood the comments of the referee here (regarding "the upper n cm that are controlled by the data depth distribution") if no standardization was implemented the baseline depth that would be uniform across all data would be just the top 1 cm - which would lead to significant information loss and is not particularly informative. We recognize the term "decay function" may be misleading here - we are not specifically measuring/estimating a rate of degradation (it was meant to refer to the expected and identified decreasing trend with depth). This term has been removed for clarity – see line 500. We also recognize the uncertainty that is introduced by this standardization process, and this is discussed within the "Model interpretation and uncertainties" section of the discussion. We also mention in this section of the manuscript that "The results from this study do however highlight, that within the top 30 cm of sediment, the spatial location of the sample is a far stronger driver of organic carbon content than the sediment sampling depth".**

*Section 2.7: I do not agree with using this data as observed data. Why not just upsample the prediction? The observed data is not real and thus should not be used in this manner. No prediction even if the predictors are fantastic will give you the correct result if the observed data is not solid. Further, why sample it spatially randomly and not at the same locations you have observed data for OC? I think you are arbitrarily making your predictions better by randomly sampling as you are covering the feature (predictor) space more uniformly. We find the same phenomenon occurs when performing this on synthetic data. The synthetic data will always outperform as it covers the feature space more uniformly and predictors are inherently tied to spatial phenomenon. I would suggest if you (although I do not agree with it) are to use this MAR data like this then you should sample it spatial and with as many samples as other datasets (e.g., OC) that have been sampled. This will represent the natural (but unfortunate) bias that occurs in marine sediment sampling.*

**RESPONSE: Agreed. As such, the MAR and CAR modelling and prediction has been removed from this study. See further details above.**

*"Line 530: This is not completely spatial explicit if you are adding x and y coordinates as predictor variables?"*

**RESPONSE: The x and y coordinates are not added as predictor variables in the models, but rather they are only included as part of the multivariate distance *plot_geodist* calculations to incorporate appropriate spatial considerations into the CV fold design. This sentence (now in Appendix A5 lines 1258-1261) now states "as the spatial distribution of data is a key consideration to ensure robust cross-validation, the x- and y-coordinates of each data point were also included in the plot_geodist calculations". The word "explicit" has been removed throughout as this may add confusion (lines 575, 628, 1033, 1060, 1065).**

Line 607: Why change the *mtry* and *min_n* number and not the number of trees?
**RESPONSE: It is relatively standard practice within random forest modelling to not tune over the trees hyperparameter and set this value to a computationally feasible large number (e.g. (Probst et al., 2019) – "The number of trees in a forest is a parameter that is not tunable in the classical sense but should be set sufficiently high"). This citation has been added to the manuscript. See line 651**

*"Line 626: How did you merge these datasets? Were there edge effects?"*
**RESPONSE: This simply allows running the computational process of prediction in serial chunks. Data were split with random sampling across the entire model domain - no edge effects were expected or apparent. Further clarification has been added within the text – see lines 677-683.**

*"What about using confidence intervals on non- normally distributed data?"*
**RESPONSE: There are a number of ways to calculate prediction uncertainties/intervals from random forest models with many arguments for and against different methods (e.g. see (Johnstone and Zhang H, 2020; Roy and Larocque, 2020; Wager et al., 2014). While we recognize that our chosen method contains assumptions regarding the distribution of the prediction error, we feel that it is a robust and justifiable choice, being the primary method implemented in the most commonly used random forest modelling packages** (Kuhn and Wickham, 2020; Wager et al., 2014; M. Wright N. and Ziegler, 2017)**.**

*"Section 2.9: Why are you calculating dry bulk density in several different ways? Why are you calculating porosity with three different equations? This section lacks clarity and I left feeling confused about what was done."*
**RESPONSE: Based on this comment, and related comments from reviewer 2, the methods used for estimating dry bulk density have been simplified (now section 2.6). Further explanation regarding the logic and reasoning behind the calculations which are used has also been added. See lines 703-735**

*"Did you do anything to avoid collinearity between the predictor grids that were selected?"*
**RESPONSE: It is generally considered that random forest model predictions are robust to issues with collinearity. Additionally, the forward feature selection process adopted in this study limits the likelihood of collinearity as it would be expected that the addition of a highly correlated predictor variable would not provide significant improvement of prediction accuracy.**

*"Figure 4 caption: Indicate more clearly which parts of these is ALE (distros drawn by line) and which is a rug plot (bottom)"*
**RESPONSE: The figure legends of the ALE plots (Figures 4 & 7) have been edited to make this distinction clearer.**

*"On what grounds (citations?) were certain predictors selected as a priori for input to the various predictions? (e.g., Fig 3)"*

**RESPONSE:** "a priori" was probably not the correct term to use here. As outlined in the methods (now Appendix A5), two variables were selected prior to the initiation of the forward feature selection process based on selecting those variables with highest variable importance within a random forest model. The term "a priori" has been changed to "initial" (see lines 808, 849, 1290) and further detail has been added to figure legends 3 & 6.

*"Line 788: is this a typo? RMSE of 0.09 arcsin{%OC} also what about line 808 "RMSE of 0.206 log10{g cm-2yr-1}." Why say it like this?"*
**RESPONSE:** We appreciate that this RMSE value is not overly intuitive, however back-transformations of RMSEs are not valid. We felt it is still important to report the RMSE even if it is somewhat unintuitive. However, more information regarding model uncertainty can be found from the $R^2$ values and prediction confidence intervals.

**RESPONSE TO REFEREE 2 - Dr Misiuk**

***General comments***

*"The Methods section is a bit long, but this may be acceptable given the scope of the project. If the authors are able to improve the conciseness of this section it would be nice for the reader, but maybe that is not possible."*
**RESPONSE:** We appreciate that the methods section was quite lengthy. We have therefore moved some of the general (or less integral) parts of the methods to an Appendix where possible – now Appendix A.

*"I am not sure if the method employed to calculate confidence intervals for various location specific carbon parameter predictions, and also for the estimates of carbon stocks across the entire Canadian margin, is appropriate. The same logic is applied in many places to derive confidence intervals for a parameter that is based on confidence intervals from a prior model prediction, which I discuss in a bit more detail below. This is my one major concern with the manuscript. Happy to be corrected if I am mistaken here, but please see my comments on the subject below."*
**RESPONSE:** We thank the reviewer for this comment and the further details in the relevant comments below. We agree that using transfer functions on, or compounds of, upper and lower bounds of 95% confidence intervals (CI), statistically does not necessarily produce a 95% confidence interval around the mean of the response which is being derived. However, based on the data we have available, the methodology we have used to derive uncertainty around our mean estimates are the best available, and are valid. Equivalent approaches to uncertainty calculation can be seen in similar seabed sediment carbon mapping studies (Diesing et al., 2017, 2023; Lee et al., 2019). We fully recognize, however, that these are not true 95% CIs. We therefore have changed the language throughout, with the term "95% confidence intervals" only being used when specifically referring to the directly derived CIs from random forest predictions of mud and OC content. All further derived layer (i.e. dry bulk density and OC density) are now referred to as "best available approximations of the lower and upper bounds of uncertainty around our mean estimates of organic carbon/dry bulk

**density" (see lines 732-735, 762-764). Further, details on these assumptions/uncertainties have also been added to the end of the "Model interpretation and uncertainties" section of the discussion (see lines 1128-1133), and we now make note to the similar approaches taken in other studies (with citations) in lines 735 & 764.**

*Specific Comments:*

*"Page 5. The colour ramp for the bathymetry is unorthodox. This is not necessarily a problem, bit it looks a bit odd. Is there any reason why dark blue is used for shallow and light yellow for deep?"*
**RESPONSE: The colour scale for the bathymetry in Figure 1 has been updated from light blue for shallow to black for deep.**

*"Page 10-11. Suggest making it clear here that "SPM" refers to suspended particulate matter. It is not formally defined."*
**RESPONSE: The definition of the acronym has now been made clear within the text. See line 256.**

*"Page 11, L 245. Consider using the word "disparity" rather than "dissimilarity" here."*
**RESPONSE: Alteration made as suggested. See line 261.**

*"Page 19. It is not currently clear to me why the MAR netCDF had to be sampled and modelled when you already had full-extent spatial predictions of MAR. Why not just resample/interpolate these to the appropriate resolution and projection rather than modelling them? They have already been modelled once in the first place – it seems like a second round of modelling may just compound the error associated with these predictions."*
**RESPONSE: As discussed in our response to Referee #1, the MAR and CAR components of this study have now been removed entirely from the manuscript.**

*"Page 20, L 543-560. I found this section a bit difficult to follow. It would help to be very consistent with the terminology here. For example, terms like "analysis data" and "assessment data" are used. Are these the same as "training" and "test" data, which are used later on? I am pretty sure these are the same thing, but consistency would be good, especially because the validation procedure was fairly involved."*
**RESPONSE: The clarity of this section (now Section 2.5) should now be improved, with more complex details regarding the CV-fold selection and the forward feature selection process moved to Appendix A5. The terminology of "analysis and assessment data" have been removed from this section (now Appendix A5) and have been explained more clearly in the manuscript (they are not the same as the test-training split; they were referring to the splits of the training data into validation sets i.e. the CV folds. As suggested by the reviewer below, the term "validation sets/data" is now used when referring to these splits).**

Comment continues.. *"Also in this paragraph, you write that "…This function creates density plots of nearest neighbour distances in multivariate predictor space between all response*

*data…" By definition, aren't nearest neighbour distances the distance between a data point and its nearest neighbour (in multivariate space), not between all data points? Relatedly, could you make it clear what these multivariate distances are? Were the predictors normalized before calculating (Euclidean) distances? That would have a very large impact on the distance calculations. This is also not clear from the appendix figures, which are just labelled "dist""*

**RESPONSE: The sentence regarding sample-sample distance between response data has been removed as it is not important when assessing appropriate CV fold structures here (see lines 1246-1261). The packages and procedures which are used here and were produced by Meyer et al. (as cited within the manuscript), do indeed normalize predictors before calculating distances. This has been added to section A5 for clarity – see line 1250. A sentence has also been added to the legends of each of the relevant appendix figures stating "dist = Multivariate Euclidean distance in predictor space after normalization of predictors."**

*"Page 21-22, L 586-605. Can you clarify how the a priori feature selection was accomplished? You write that variable importance was calculated using a "basic random forest model with all training data and predictor variables…" Does this imply that you used the out-of-bag samples to estimate variable importance, or was it estimated on the training data?"*

**RESPONSE: Variable importance estimates were indeed estimated using out-of-bag samples. This has been added to the text and a citation to** (M. N. Wright et al., 2016) **has been added so further details on the method built into the *ranger* package can be sought by the reader if required. – see lines 1288-1297.**

*"Page 22. I found this section also hard to follow, and again, believe it would be helpful if the terminology was consistent. I'm assuming the "training" and "test" sets here are the "analysis" and "assessment" data from above? This may not be obvious to some readers because "assessment" or "validation" data are often used for model tuning, while a separate "test" set is withheld to calculate the final accuracy. I'm having trouble understanding the hyper-parameter optimization and validation design described on this page. Can you confirm that separate models were built and tuned for each fold, and were also validated using the same design, and that the out-of-bag (OOB) samples were not used anywhere in this procedure? If that is the case, why not use the OOB samples to tune hyper- parameters, then use the folds for validation? Tuning hyper-parameters on the validation data can slowly lead to a form of overfitting."*

**RESPONSE: We apologize for the confusion. As discussed in our response to this reviewer above (regarding Page 20, L 543-560), the terminology regarding "training", "testing" and "analysis and assessment data" has been simplified, and the term "validation data" has been added when discussing the CV folds as suggested by the reviewer. Edits have been made to this section more generally to aid clarity – see Section 2.5 (lines 615-659) and Appendix A5 (lines 1245-1299). The referee is correct that out-of-bag samples/errors were not used to assess hyperparameters or overall model validation. We follow the procedure as adopted within the *tidymodels* package (Kuhn and Silge, 2023; Kuhn and Wickham, 2020) which requires definition of the structure of data-splitting for tuning and validation as this allows for more control of model metric calculations. i.e. using out-of-bag error for model validation and tuning would be similar to a random cross-validation/assessment approach, which as we discuss in this manuscript, is not appropriate for this modeling exercise.**

*"Page 22, L 615. Sorry, but in the same vein the terminology is confusing to me here. You write that: "Overall model performance metrics (RMSE and R2) were then calculated using the predictions across all CV folds with optimal hyperparameters and the last-fit; while predictor variable importance was calculated by fitting an additional model across all training data using optimal tuning parameters and the importance calculated through permutation." Does this imply that, while performance was calculated using CV, the variable importance was calculated using the training data (i.e., the "analysis" data)? When you say permutation, does this mean you permute the predictor variables and measure the impact of the model predictions on the training fit then? Or are the OOB data being used here? How were these "optimal tuning parameters" selected if hyper-parameter tuning was performed per fold?"*

**RESPONSE: Again we apologize for the confusion. This section (see lines 656-701) has been edited to increase overall clarity. The reviewer is correct that performance was calculated using the CV fold structure, but variable importance was calculated on overall training data and permutation of variable values with measurement of impact on out-of-bag data, following Kuhn and Silge, 2023; Kuhn and Wickham, 2020; Wright et al., 2016. To clarify these points, we now state in the manuscript "The performance of each of the 11 hyperparameter combinations was assessed by calculating the root mean squared error (RMSE) on predictions of the validation data across all CV folds, with the optimal hyperparameter combination selected as that with the lowest RMSE (Meyer et al., 2019, 2023)." See lines 656-659.**

*"Page 22. These paragraphs appear to imply that you predicted each of the 10 models from the CV across the entire domain, correct? Or was it a single model fitted "across all training data" from above? In either case, what does it mean that the data were randomly split into 150 samples, predicted, then merged to produce the final raster layer? What data were split and why is it random? Please clarify this."*

**RESPONSE: As stated in the manuscript "After selection of the best performing hyperparameter combination, a single last-fit model was constructed on the entire training set" and "predicted values were calculated across the entire model domain using the last-fit model and the predictor variable raster stack (Kuhn and Silge, 2023; Kuhn and Wickham, 2020)" – see lines 660-661 & 673-674. To improve clarity, text regarding the splitting of the predictor variable raster stack into 150 partitions for final model predictions has been further developed – see lines 677-683.**

*"Page 23, L 658. "Fourth", not "forth"."*

**RESPONSE: This section has been widely altered so this typo is no longer present. – see line 735**

*"Page 24, L 675-678. Does this mean that the final bulk density estimates were the mean of the 14 individual estimates? How did you get standard errors after aggregating these predictions? Did you just use the standard deviation of the 14 predictions? How then can you get a 95% confidence interval? This needs clear explanation. As far as I can tell, you have a 95% CI for four of the transfer functions… how could you possibly pool these, in addition to the other estimates?"*

**RESPONSE: Based on this comment, and similar comments from Reviewer 1, the methodology for estimating mean dry bulk density and the uncertainty bounds has been simplified (now Section 2.6, lines 703 - 735). Dry bulk density is now only estimated based on a relationship to predicted mud content, with the uncertainty bounds calculated from the estimated mud content CI.**

*"Page 24, L 683-685. You use the 95% CI bounds of the %OC predictions and the MAR predictions to calculate a 95% CI for OC accumulation rates. Why is that appropriate? For example, the upper bound of the %OC and MAR intervals implies you are 97.5% sure the mean value at a given location should be less than that value. Your calculation implies that, given 97.5% confidence that the mean OC% is < x, and given 97.5% confidence that the mean MAR is < y, you are 97.5% sure that the mean OC accumulation rate is < xy. It may be worthwhile to consult a statistician here, but I don't think that's the proper way to calculate a confidence interval. Would you not need some information about the joint distribution of these variables? I'm not a statistician; if I am wrong, please provide some source for this approach. This has very important implications for your calculations across the entire model domain. Is it appropriate to take the upper and lower confidence bounds of multiple variables, multiply them together, and aggregate those to arrive at 95% confidence that the total OC content across the entire shelf is within a certain range?"*
**RESPONSE: See overall response in general comments.**

*Page 26 and surrounding. A minor comment, but it would be nice if the fonts of all plot elements (e.g., axis labels) were consistent across figures.*
**RESPONSE: Font sizes updated for consistency – See Figures 3,4,6 and 7**

*"Page 28. The high importance of the predicted mud content raises an important question for me regarding the potential for "data leakage" in your validation. Can you confirm that none of the %OC measurements came from data points where the mud content was also measured and used for those models? It seems possible that this could have occurred since many of both data points were sourced from NRCan, and that there were many more mud samples than %OC in your dataset. If the mud data points were used to create map predictions at all locations where you have %OC measurements, yet the %OC measurements were sourced from data points that occur in the mud dataset, it is possible that the test data from your cross-validation folds are "leaking" information to the training data via the predicted mud layer. This would occur due to expected correlations between the mud and organic carbon content if they were obtained from the same sample, and not assigned the same CV folds. It would potentially inflate both the estimates of importance of the mud layer, and the estimates of model performance, despite your substantial efforts at ensuring an appropriate CV design."*
**RESPONSE: We have tested for overlap and found just 31 out of 2356 %OC samples (1.3%) directly spatially overlap with the location of mud content was measured. We therefore feel that this would not cause significant data leakage or impact variable importance or model performance. Text regarding this comment has been added - see lines 691-694.**

*"Page 32. Similar comment to previous. You are assuming that the 95% CI can be acquired for*

*dry bulk density predictions simply by performing the calculations using upper and lower bounds of the mud predictions. I'm not sure that is the case. I think it is true to say that you are calculating the likely dry bulk density given the 95% CI bounds of the mud predictions, but I don't think that is the same as calculating the 95% CI of the dry bulk density."*
**RESPONSE: See overall response in general comments.**

*Page 35. Same comments as previous regarding the propagation of confidence intervals.*
**RESPONSE: See overall response in general comments.**

**REFERENCES WITHIN RESPONSES**

Diesing, M., Kroger, S., Parker, R., Jenkins, C., Mason, C., and Weston, K. (2017). Predicting the standing stock of organic carbon in surface sediments of the North-West European continental shelf. *Biogeochemistry*, *135*(1), 183–200. https://doi.org/10.1007/s10533-017-0310-4

Diesing, M., Paradis, S., Jensen, H., Thorsnes, T., Bjarnadóttir, L. R., and Knies, J. (2023). Organic Carbon Stocks and Accumulation Rates in Surface Sediments of the Norwegian Continental Margin. *ESS Open Archive*, pre-print.

Gregr, E. J., Haggarty, D. R., Davies, S. C., Fields, C., and Lessard, J. (2021). Comprehensive marine substrate classification applied to Canada's Pacific shelf. *PLOS ONE*, *16*(10), 1–28. https://doi.org/10.1371/journal.pone.0259156

IPPC. (2019). 2019 Refinement to the 2006 IPCC Guidelines for National Greenhouse Gas Inventories. *The International Plant Protection Convention*.

Johnstone, C., and Zhang H, H. (2020). PiRF: Prediction Intervals for Random Forests. *R Package Version 0.1.0,* https://CRAN.R-project.org/package=piRF.

Kuhn, M., and Silge, J. (2023). *Tidy Modeling with R: Vol. Version 1.0.0*. O'Reilly Media, Inc.

Kuhn, M., and Wickham, H. (2020). *Tidymodels: A collection of packages for modeling and machine learning using tidyverse principles*. https://www.tidymodels.org

Lee, T. R., Wood, W. T., and Phrampus, B. J. (2019). A Machine Learning (kNN) Approach to Predicting Global Seafloor Total Organic Carbon. *Global Biogeochemical Cycles*, *33*(1), 37–46. https://doi.org/10.1029/2018gb005992

Maxwell, A. E., and Shobe, C. M. (2022). Land-surface parameters for spatial predictive mapping and modeling. *Earth-Science Reviews*, *226*, 103944. https://doi.org/10.1016/j.earscirev.2022.103944

Meyer, H., Milà, C., and Ludwig, M. (2023). CAST: 'caret' Applications for Spatial-Temporal Models. *R Package Version 0.7.1*. https://CRAN.R-project.org/package=CAST

Meyer, H., Reudenbach, C., Wöllauer, S., and Nauss, T. (2019). Importance of spatial predictor variable selection in machine learning applications – Moving from data reproduction to spatial prediction. *Ecological Modelling*, *411*, 108815. https://doi.org/10.1016/j.ecolmodel.2019.108815

Mitchell, P. J., Aldridge, J., and Diesing, M. (2019). Legacy Data: How Decades of Seabed Sampling Can Produce Robust Predictions and Versatile Products. *Geosciences*, *9*(4). https://doi.org/10.3390/geosciences9040182

Probst, P., Wright, M. N., and Boulesteix, A.-L. (2019). Hyperparameters and tuning strategies for random forest. *WIREs Data Mining and Knowledge Discovery*, *9*(3), e1301.

https://doi.org/10.1002/widm.1301

Roy, M.-H., and Larocque, D. (2020). Prediction intervals with random forests. *Statistical Methods in Medical Research*, *29*(1), 205–229. https://doi.org/10.1177/0962280219829885

VERRA. (2020). Methods for Monitoring of Carbon Stock Changes and Greenhouse Gas Emissions and Removals in Tidal Wetland Restoration and Conservation Project Activities (M-TW). *VCS Module VMD0051, Sectoiral Scope 14*, *1*.

Wager, S., Hastie, T., and Efron, B. (2014). Confidence Intervals for Random Forests: The Jackknife and the Infinitesimal Jackknife. *Journal of Machine Learning Research*, *15*(48), 1625–1651.

Wright, M., N., and Ziegler, A. (2017). {ranger}: A Fast Implementation of Random Forests for High Dimensional Data in {C++} and {R}. *Journal of Statistical Software*, *77*(1), 1–17.

Wright, M. N., Ziegler, A., and König, I. R. (2016). Do little interactions get lost in dark random forests? *BMC Bioinformatics*, *17*, 1–10.